**Photochemical age of air pollutants, ozone, and secondary organic aerosol in**
**transboundary air observed on Fukue Island, Nagasaki, Japan**
Satoshi Irei,[1][†]Akinori Takami,[1] Yasuhiro Sadanaga,[2] Susumu Nozoe,[1][‡] Seiichiro Yonemura,[3]
Hiroshi Bandow,[2] and Yoko Yokouchi[1]
[1]National Institute for Environmental Studies, 16-2 Onogawa, Tsukuba, Ibaraki 305-8506,
Japan
[2]Department of Applied Chemistry, Graduate School of Engineering, Osaka Prefecture
University, 1-1 Gakuencho, Naka-ku, Sakai, Osaka 599-8531, Japan
[3]National Institute for Agro-Environmental Sciences, 3-1-3 Kannondai, Tsukuba, Irabaki 305-
8604, Japan.
[†]Present address: Department of Biology, Chemistry, and Marine Science, University of the
Ryukyus, 1 Senbaru, Nishihara, Okinawa 903-0213, Japan.
[‡]Present address: National Museum of Emerging Science and Innovation, Aomi 2-3-6, Koto,
Tokyo 135-0064, Japan.
**Abstract**
To better understand the secondary air pollution in transboundary air over westernmost Japan,
ground-based field measurements of the chemical composition of fine particulate matter ($\leq$1
μm), mixing ratios of trace gases species (CO, $O_3$, $NO_x$, $NO_y$, *i*-pentane, toluene, and ethyne),
and meteorological elements were conducted with a suite of instrumentation. The CO mixing
ratio dependence on wind direction showed that there was no significant influence from
primary emission sources near the monitoring site, indicating long- and/or mid-range transport
of the measured chemical species. Despite the considerably different atmospheric lifetimes of
$NO_y$ and CO, these mixing ratios were correlated ($r^2 = 0.67$). The photochemical age of the
pollutants, $t$[OH] (the reaction time $\times$ the mean concentration of OH radical during the
atmospheric transport), was calculated from both the $NO_x/NO_y$ concentration ratio ($NO_x/NO_y$
clock) and the toluene/ethyne concentration ratio (hydrocarbon clock). It was found that the
toluene/ethyne concentration ratio was significantly influenced by dilution with background
air containing 0.16 ppbv of ethyne, causing significant bias in the estimation of $t$[OH]. In
contrast, the influence of the reaction of $NO_x$ with $O_3$, a potentially biasing reaction channel on
$[NO_x]/[NO_y]$, was small. The $t$[OH] values obtained with the $NO_x/NO_y$ clock ranged from 2.9
$\times 10^5$ to $1.3 \times 10^8$ h molecule $cm^{-3}$ and were compared with the fractional contribution of the
$m/z$ 44 signal to the total signal in the organic aerosol mass spectra ($f_{44}$, a quantitative oxidation
indicator of carboxylic acids) and $O_3$ mixing ratio. The comparison of $t$[OH] with $f_{44}$ showed
evidence for a systematic increase of $f_{44}$ as $t$[OH] increased, an indication of SOA formation.
To a first approximation, the $f_{44}$ increase rate was $(1.05 \pm 0.03) \times 10^{-9} \times$ [OH] $h^{-1}$, which is
comparable to the background-corrected increase rate observed during the New England Air

Quality Study in summer 2002. The similarity may imply the production of similar SOA component, possibly humic-like substances. Meanwhile, the comparison of $t$[OH] with $O_3$ mixing ratio showed that there was a strong proportional relationship between $O_3$ mixing ratio and $t$[OH]. A first approximation gave the increasing rate and background mixing ratio of ozone as $(3.48 \pm 0.06) \times 10^{-7} \times$ [OH] ppbv h$^{-1}$ and 30.7 ppbv, respectively. The information given here can be used for prediction of secondary pollution magnitude in the outflow from the Asian continent.

*Key words:SOA; ozone; photooxidation of air pollutants; long-range transport; transboundary air pollution; East Asia*

## 1. Introduction

During the last decade, the dramatic growth of the Chinese economy has increased emission of air pollutants such as volatile organic compounds, particulate matter (PM), and nitrogen oxides ($NO_x$), which is the sum of nitrogen monoxide (NO) and nitrogen dioxide ($NO_2$). In northeast Asia, air masses generally move from west to east, and therefore pollutants emitted on continental China are frequently carried to Japan. The influence of air pollution is becoming severe in rural areas of westernmost Japan, such as Fukue Island. Atmospheric oxidation of primary pollutants produces secondary pollutants, such as ozone ($O_3$), secondary particulate

organic matter (also known as secondary organic aerosol or SOA), which is formed by
oxidation of volatile organic precursors. A better understanding of these secondary pollutants
is important not only for purely scientific reasons but because such pollutants are a matter of
great public concern. SOA is one of the least understood subjects in atmospheric chemistry
(Ebben *et al.*, 2014), despite the fact that it has been studied extensively owing to its potential
adverse effects on human health and its role in cloud condensation. Although state-of-the-art
techniques, such as aerosol mass spectrometry, have substantially improved our understanding
of SOA (Zhang *et al.*, 2005; Jimenez *et al.*, 2009), many questions about SOA still remain, such
as its constituents, production mechanisms, and fates.

To understand SOA, we must evaluate the progress of the chemical reactions of its

constituents. The progress of photochemical reactions in the atmosphere has frequently been
evaluated in terms of a "photochemical age," designated $t$[OH], which can be derived from
non-methane hydrocarbon (NMHC) ratios (Roberts *et al.*, 1984; Rudolph and Johnen, 1990)
and from $NO_x$ ratio to total odd nitrogen ($NO_y$) (Parrish *et al.*, 1992). Recent field studies
combining aerosol mass spectrometry measurements and determination of $t$[OH] have
provided new information about photochemically produced SOA (de Gouw *et al.*, 2005;
Takegawa e*t al.*, 2006; Kleinman *et al.*, 2007; Liggio *et al.*, 2010). Our previous field studies
conducted on Fukue Island in Japan demonstrated a systematic trend for the fractions of
carboxylate in the organic aerosol ($f_{44}$) with $t$[OH], evidence of SOA production (Irei *et al.*,
2014). However, the study period was short (only 10 days), and a longer observation period is
necessary to obtain more-convincing evidence of SOA production. Furthermore, inconsistent
results regarding the association between SOA production and $t$[OH] were observed at the same
location during a different time period (Irei *et al.*, 2015). The study described in this paper is
an extension of our previous studies, and the objective was to deepen our understanding of the
association between oxidation products (SOA and $O_3$) and $t$[OH] in transboundary air.

**2.  Experimental**
Field measurements were conducted from December 2010 to May 2011 at the Fukue
atmospheric monitoring station (32.8°N, 128.7°E), a rural site on the northwestern peninsula
of Fukue Island, Nagasaki Prefecture, Japan (Figure 1). As mentioned earlier, data collected
during a 10 day observation period in December 2010 have already been reported, and the
reported results are a part of data in this study. The monitoring station is ~1 km away from the
residential area of the peninsula and is ~60 m higher in altitude. The site is located in-between
small pastures. Possible sources of anthropogenic emissions of fine aerosol and trace gas
species include agricultural waste burning, home incinerators, automobiles occasionally
passing by the station, and tractors sometimes mowing the pastures. For all the measurements
the ambient air was measured or sampled 1 ~ 3m above the rooftop of the station (3 ~ 5m height
from the ground). An independent sampling line was assembled for each chemical species
measurement. The ambient air was sampled at 1 L min$^{-1}$ through 5 m × quarter-inch o.d. PTFE
tubing for the CO and O$_3$ measurements and at 0.5 L min$^{-1}$ through the same type of tubing for
the NO$_x$ and NO$_y$ measurements, respectively. A molybdenum converter for the NO$_y$
measurements was set at the inlet of sampling line. For the particle and NMHC measurements,
the ambient air was suctioned at 3 L min$^{-1}$ and 5 L min$^{-1}$ at the first stage through the sampling
lines of ~4 m × half-inch o.d. and ~10 m × five-fifth-inch o.d. stainless steel tubing (GL Science,
Japan), respectively. The measurements were then made by sampling a part of the flowing air.
For the particle measurements only, a PM$_{2.5}$ cyclone separator (URG 2000-30ED, URG Corp.
Chapel Hill, NC, USA.) was attached to the inlet of the sampling line to cut off particles larger
than PM$_{2.5}$.

The 10 min average chemical composition of fine aerosol (~PM$_{1.0}$) was measured with

an Aerodyne quadrupole aerosol mass spectrometer (AMS, Aerodyne Research Inc., Billerica,
MA, USA). Details of the instrumentation and the method for determination of chemical
species concentrations are described elsewhere (Jayne *et al.*, 2000; Allan *et al.*, 2004). The
AMS was calibrated approximately once a month with 350 nm dried ammonium nitrate
particles for determination of ionization efficiencies. The temperature of the flash vaporizer
was set to 873 K during the field measurements and calibration measurements. A collection
efficiency of 0.74 was used for determination of chemical species concentrations; this value
was determined from comparison between sulfate concentrations measured by means of AMS
and non-sea-salt sulfate concentrations determined by means of total suspended particulate
filter sample analysis during the field study in December 2010 (Irei *et al.*, 2014). The detection
limits (DLs) of the mass spectrometer for chloride, nitrate, ammonium, sulfate, organics, $m/z$
43 (an indicator for detection of hydrocarbon and aldehyde) , and $m/z$ 44 (an indicator for
detection of carboxylic acid) were determined by $3 \times$ standard deviation (SD) of blank
concentrations obtained by measuring filtered ambient air (HEPA Capsule, Pall Corp.) for 2 ~
16 hours. The blank measurements were conducted every month during the study period. The
average DLs of these species were 0.02, 0.04, 0.2, 0.4, 0.5, 0.02, and 0.06 $\mu g\ m^{-3}$, respectively.

Mixing ratios for $NO_x$ and $NO_y$ were measured in situ to retrieve the $t$[OH], an indicator

of atmospheric oxidation. Measurement methods for $NO_x$ and $NO_y$ mixing ratios were
developed with an LED converter and a molybdenum converter, respectively, together with
commercially available $NO_x$ analyzers (Model 42 i-TL, Thermo Scientific). These instruments
are described in detail elsewhere (Sadanaga *et al.*, 2010; Yuba *et al.*, 2010). DLs for NO, $NO_2$,
and $NO_y$ were about 0.06 ppbv with 1 min averaging time. Mixing ratios of CO and $O_3$ were
measured in situ with a CO analyzer (Model 48, Thermo Scientific) and an $O_3$ analyzer (Model
49i, Thermo Scientific), respectively. The DLs of these instruments were 10 and 5 ppbv with
10 and 1 min averaging time, respectively. The analog signal output for these trace gas species
was recorded every second using a data logger (NR-1000, KEYENCE), and hourly average
mixing ratios were used for data analysis. Selected NMHCs (ethyne, *i*-pentane, and toluene)
were also measured hourly with a gas chromatograph equipped with a flame ionization detector
(6890N, Agilent Technologies) and coupled with an automated cryo preconcentration sampler
(Yokouchi, 2008). Ethyne, *i*-pentane, and toluene were chosen because those can be used as
markers for vehicular emissions (Tang *et al.*, 2009; Wang *et al.*, 2015). The choice of toluene
was also owing to one of the possible precursors of atmospheric SOA (Grosjean and Seinfeld,
1989; Seinfeld and Pandis, 1999). The volatile organic compounds in 600 mL of ambient air
were collected cryogenically from the main stream of the previously referred sampling line at
a flow rate of 40 mL min$^{-1}$ (*i.e.*, a 15 min sampling period for a single measurement). Target
compounds were identified and quantified on the basis of comparison with retention times and
peak area counts for standards; specifically, a standard gas containing 1 ppb of each target
compound was analyzed once a day. The DLs for ethyne, *i*-pentane, and toluene were 2.5, 1.5,
and 1.5 pptv, respectively.
Additionally, ambient temperature, relative humidity (RH), precipitation, and wind
speed and direction were measured with a weather transmitter (WXT 520, VAISALA, Helsinki,
Finland).

**3. Results and Discussion**
**3.1. Meteorological observations**
Measured ambient temperature ranged from 274.6 to 301.2 K, the mean ± SD = 286.6 ± 5.4 K
(Figure S-1). Ambient temperature showed a clear seasonal variation, and a polynomial best fit
curve $\pm$ 5 K covered ~90% of the data points and reproduced the observed trend.

Precipitation events were observed occasionally (Figure S-1), but their frequency and

strength did not seem to significantly affect our overall interpretation of the entire data set.
Therefore, in the analyses described hereafter, we included the data collected during the
precipitation events, unless otherwise noted. RH varied between 25% and 100% and seemed
to be relatively constant from December to February and to vary more widely from March to
May (Figure S-1).

A polar plot of hourly average wind speed shows that it ranged from 0.2 to 10 m s$^{-1}$

(Figure S-2). The mean $\pm$ SD of wind speeds during the observation period was $3 \pm 1$ m s$^{-1}$,
and the 90th, 25th, and 10th percentile cut-off values were 4, 2, and 1 m s$^{-1}$, respectively. This
information suggests that medium-strength winds (*i.e*., wind speeds of 2–4 m s$^{-1}$) blew most of
the time during the study period. Because wind directions measured at wind speeds of <1 m s$^{-}$
$^{1}$ are often treated as invalid, the fact that the 10th percentile cut-off for our data was 1 m s$^{-1}$
indicates that 90% of our wind direction data were valid. The most prevalent wind directions
were between northwesterly and northeasterly (35%) and between northeasterly and
southeasterly (26%). The prevalence of wind from the residential area of the peninsula (from
the direction between southeasterly and southwesterly) was about 17%.

**3.2. Chemical species concentrations**

The results of statistical analysis of the concentrations of chemical species in fine PM are summarized in Table 1, along with the results for gas-phase species. Because sea-salt PM tends to be coarse, the very low concentrations of chloride measured by means of AMS indicate that most of the chloride originating from sea salt was eliminated at the AMS inlet, which selects for fine PM. The mean concentrations ± SDs of the chemical species in fine PM were similar to those observed in 2003 at the same location (Takami *et al.*, 2005) and at Cape Hedo, Okinawa (Takami *et al.*, 2007). Sulfate was the predominant chemical species in fine PM throughout the observation period, accounting for 46% on average, and was followed by organics (29%), ammonium (16%), and nitrate (8.0%). The concentrations of nitrate, the detection of which is often an indication of the proximity of its emission source, were high in this study even though the monitoring station was located in a rural area. In many cases, the amount of nitrate in fine PM decreases or shifts to larger PM during long-range transport (Takiguchi *et al.*, 2008, and references therein). Because there are no large emission sources of primary nitrate around the monitoring station, the high nitrate concentrations probably indicate mid- or long-range transport of pollutants from locations off the island. Temporal variation of the concentrations of organics in fine PM measured by means of AMS showed no seasonal trend, but some high-concentration episodes were observed (Figure S-3). It was also found that the concentrations of organic aerosols in the study period from 6 to 16 December, which was previously reported

(Irei *et al.*, 2014), were relatively low. In the time-series plot, the $f_{44}$, the fraction of $m/z$ 44 in
the organic mass spectrum or the fraction of carboxylate in organic aerosol, seemed to rise from
~0.12 to ~0.15 around the end of March. This increase may have been owing to greater
production of oxygenated organic compounds in spring than in winter because of the increasing
sunlight irradiance in the spring, which was indicated by the times-series plot of ambient
temperature (Figure S-1).

Most of the $O_3$ mixing ratios were <55 ppbv, and the mean of 45 ppbv was consistent

with the annual mean of ~50 ppbv observed at the same location in 2011 (Kanaya *et al.*, 2016);
this annual mean of ~50 ppbv was the lowest annual mean $O_3$ mixing ratio observed over the
course of 6 years (2009 − 2014) at this location. A times-series plot of hourly average $O_3$
mixing ratios showed that although there were some episodes of high mixing ratios, the mixing
ratios seemed to vary between ~25 and ~50 ppbv from December to February and then were
prone to gradually increase from the beginning of March to May (Figure S-4a). Similar
seasonal trends have been observed at the same location (Kanaya *et al.*, 2016) and at other
remote sites in East Asia (Pochanart *et al.*, 2002; Suthawaree *et al.*, 2008; Kanaya *et al.*, 2016,
references therein). This trend was similar to the $f_{44}$ trend described above and therefore can
also be explained in terms of an increase in sunlight irradiance to polluted air masses
transported from the Asian continent. Meanwhile, according to the observations at the other
remote sites referred above, the $O_3$ mixing ratios tend to drop starting in May and continuing
into the summer because the origin of air masses changes from the continent directly to the
Pacific Ocean; the oceanic air masses generally contain much lower quantities of $O_3$ and its
precursors. The drop in the $O_3$ mixing ratios observed between May 9 and May 12 was
compatible with the influence of the oceanic air masses demonstrated by the back trajectories
of air masses (Figure S-5) modeled by HYSPLIT (Draxler and Rolph, 2013).
The $NO_x$ mixing ratios ranged from lower than the DL (LDL, < 0.006 ppbv) to 12.70
ppbv (mean ± SD = 1.39 ± 1.16 ppbv), and the $NO_y$ mixing ratios ranged from 0.13 to 25.41
ppbv (mean ± SD = 4.86 ± 3.49 ppbv). The upper quartile cut-offs for these mixing ratios were
1.70 and 6.03 ppbv, respectively. NO was found to be the very minor component of $NO_x$. The
median and lower and upper quartile cut-off values of NO were LDL, LDL, and 0.06 ppbv,
respectively. Compared to the mixing ratios observed in other field studies (Pandey Deolal *et*
*al.*, 2012, and references therein), most of these mixing ratios fell between those observed at
European rural and background sites. No time-dependent trend was observed for the $NO_x$ or
$NO_y$ mixing ratio (Figure S-4b,c). Episodes of high mixing ratios were observed irregularly.
The CO mixing ratios ranged from 57 to 1136 ppbv, and the median, upper, and lower
quartile cut-off values were 204, 272, and 160 ppbv, respectively; no seasonal trend was
observed (Figure S-4d). Except for some episodes of high mixing ratios, the observed mixing
ratios below the upper quartile cut-off seem to be comparable in magnitude to those observed
from 2002 to 2005 at various rural and remote locations in the region of the East China Sea
(Suthawaree *et al.*, 2008; Tanimoto *et al.*, 2008), indicating that the mixing ratios we observed
reflected the background mixing ratios in this region. A polar plot of the wind-sector
dependence of the CO mixing ratio showed almost no sharp increases attributable to local
anthropogenic emissions (Figure S-6). The episodes of high mixing ratios that occurred at
irregular intervals were attributed to mid-range transport of anthropogenic emissions.

To determine whether these episodes were owing to combustion-related pollution

transported from the Asian continent, we chose seven time periods with high CO mixing ratios
that lasted for more than 24 h, and we checked the back trajectories of the air masses modeled
by HYSPLIT. These episodes are listed in Table S-1, together with confirmation of
concentration rises of other chemical species during the high-CO episodes. Back trajectories
for each episode showed that the air masses were transported from the region of east-coast of
China or of west coast of Korea during these episodes. The trajectories also showed that the
episodes ended with the arrival of air masses from the Pacific Ocean or Mongolia with greater
wind speed (Figures S-7 to S-13). Thus, these results roughly support the proposition that at
least these seven high-concentration episodes were derived from the Asian continent.

Most of the observed mixing ratios for *i*-pentane, toluene, and ethyne (Table 1) were

slightly higher than the ratios observed at Cape Hedo, Okinawa, in 2000 (Kato *et al.*, 2004).
This result is consistent with the fact that pollutants transported from the Asian continent to
Fukue are often fresher than those transported to Cape Hedo (Takami *et al.*, 2007). Times-series
plots of the mixing ratios for these NMHCs showed no seasonal trends (Figure S-4e–g). The
observed sharp rises in mixing ratios of *i*-pentane, toluene, and ethyne—which lasted no more
than a few hours, indicating the influence of anthropogenic emissions near the site—accounted
for only a small portion of the observed data.

**3.3. Correlations between the concentrations of various chemical species**
Investigation of the correlations between the concentrations of various chemical species
showed that CO concentration was correlated with the concentrations of $NO_y$ ($r^2 = 0.674$),
ethyne ($r^2 = 0.724$), and organic aerosols ($r^2 = 0.562$) (Table 2). Ethyne is a combustion marker
and often originates from vehicular emission, which is one of the major sources of $NO_x$ as well.
The atmospheric lifetimes of CO and ethyne are usually determined by the reactions with OH
radicals (the most powerful oxidant in the air). Under an average OH concentration of $5 \times 10^5$
molecules $cm^{-3}$, which is the calculated diurnally averaged OH concentration during the
PEACE-A aircraft campaign over Japan in January 2002 (Takegawa *et al.*, 2004), their lifetimes
are approximately 100 and 35 days, respectively. Meanwhile, they observed very high
correlation between $NO_y$ and CO (or $CO_2$) and found that the lifetime of $NO_y$ during the long-
range transport of $NO_y$ from the Asian continent to Japan was 1.3 ~ 2.0 days, which was mainly
owing to the wet and/or dry depositions of $HNO_3$. That is, ~60% of $NO_y$ sinks within 2.0 days.
This sink is likely owing to the dry deposition because of the constant life time over the 6
different flight studies at various altitudes (0.2 ~ 4.0 km): If the sink were owing to the wet
deposition, the larger variation in the $NO_y$ lifetime should have been observed. We expect that
the order of $NO_y$ lifetime in our study is similar because the dry deposition was likely the major
sink of $NO_y$ during our study and also because the transport time of transboundary air pollutants
during our study was similar. The slope of the linear regression drawn for the $NO_y$ mixing ratio
as a function of CO mixing ratio was approximately 0.03, which is on the similar order of the
value of ~0.038 observed by Takegawa *et al.* in the 2~3 day aged plume originated from Japan.
The slope also coincided with the calculated $NO_y$/CO ratio of 0.03 in an air mass transported a
long distance from its origin to Korea using a recent emission inventory (Kim *et al.*, 2012). In
contrast, $NO_x$ to CO ratios at emission are generally higher than those values (Parrish *et al.*,
2002). Kurokawa *et al.* (2013) reported that emission ratios of $NO_x$ to CO from coal
combustion used in industry in China, which is suspected to be one of the major sources of
these pollutants observed in our study, were 0.06 ~ 0.07. With consideration of the $NO_y$ lifetime
by the depositions and of the transport time of roughly 1 ~ 3 day estimated by the back
trajectories previously referred, the discrepancy between the $NO_y$/CO observed at Fukue and
the $NO_x$/CO at emission seems to be reasonably explained by the depositional sink during the
transport. The higher coefficient of determination between CO and ethyne than that between
CO and $NO_y$ also supports the association of their correlation with their lifetimes. Despite such
significant depositional loss of $NO_y$, the positive correlation with the $r^2$ of 0.674 between CO
and NO$_y$ implies that the wet deposition, which is highly variable and influential, did not
significantly contribute to the NO$_y$ sink, in turn, the major sink of NO$_y$ was the dry deposition
depending on the gravitational residence time.
Particulate ammonium was correlated with particulate acidic components, such as
sulfate, nitrate, organics, and m/z 44 of organics. The highest correlation with *m/z* 44 ($r^2 =$
0.755) suggests that the organics were primarily composed of carboxylic acids. The observed
correlations imply that sufficient amount of ammonium was available in the gas-phase to
neutralize all these acidic components. Although it is not shown, slopes of linear regressions
between ammonium (*x*-axis) and sulfate, nitrate, or organics (*y*-axis) was 1.7, 0.74, and 1.0,
respectively. With respect to molar ratio to ammonium, sulfate and nitrate were calculated to
be 0.32 and 0.21, respectively. If only sulfate and nitrate were neutralized by ammonium, the
sum of the nitrate molar ratio and two times of the sulfate molar ratio must be equivalent to
one. Actual number for the sum is 0.85, lower than the neutralization ratio. That is, this suggests
that the amount of ammonium in PM$_{1.0}$ was more than enough to neutralize sulfate and nitrate.
Because organics was more highly correlated with ammonium than with sulfate and nitrate
(Table 2), it is feasible to explain that the excess amount of ammonium was to neutralize
organic acid. Given that all three acidic species were neutralized by ammonium, the molar ratio
of organic acid to ammonium accounts for 0.15. As the number of carboxylic group in the
organic acid molecule is referred to as *n*, the organic acid molar ratio allows us to estimate the
average molecular weight of organic acid as $120 \times n$ g mol$^{-1}$.

The overall correlation between $m/z$ 43 and $m/z$ 44 in the organic mass spectra obtained

by AMS was 0.640, but a plot of $m/z$ 43 versus $m/z$ 44 showed two distinct trends: a trend with
an $m/z$ 44 to $m/z$ 43 ratio of ~2.5 and another with a ratio of ~1 (Figure S-14), the latter of
which was clearly observed in the period from the end of December to the beginning of
February. These results suggest that two types of organic species gave fragment ions that
contributed to the $m/z$ 44 to $m/z$ 43 ratio. These species will be discussed in detail in Sect. 3.4.

**3.4. Oxidation state of organic aerosols**
As we did in previous reports for the field studies in December 2010 (Irei *et al.*, 2014) and in
March 2012 (Irei *et al.*, 2015), here we briefly discuss the results of evaluation of the oxidation
state of the organic aerosols observed during the half-year period of this study. First, we applied
positive matrix factorization (PMF) analysis to the organic aerosol mass spectra to deconvolute
the types of organic aerosols (Zhang *et al.*, 2005; Ulbrich *et al.*, 2009), and then we determined
the oxidation state of each type of organic aerosol by plotting the fractions of $m/z$ 43 ($f_{43}$) and
$m/z$ 44 ($f_{44}$) in the organic mass spectra, according to the method described by Ng *et al.* (2010).
Furthermore, we determined the mass to carbon ratios (OM/OC ratios) of the types of organic
aerosol using the method described by Zhang *et al.* (2005) to characterize the species of the
organic aerosols.
With respect to the mass spectral pattern, PMF analysis on whole dataset of organic
mass spectra gave the most feasible solution with two types of organic aerosols (Figure 2). The
mass spectral patterns of these two types of aerosols agreed well with those of hydrocarbon-
like organic aerosol (HOA) and low-volatility oxygenated organic aerosol (LV-OOA) found in
the December study ($r^2$ of 0.98 and 0.98, respectively). The patterns also agreed reasonably
with the reference mass spectra for LV-OOA and HOA in the AMS spectral database ($r^2$ of 0.94
and 0.53, respectively) made available by Ulbrich *et al.* (http://cires.colorado.edu/jimenez-
group/AMSsd/). For the identification of HOA, the negligibly small intensity at *m/z* 44 with
the high intensity at *m/z* 43 in the mass spectra was crucial to differentiate from the mass spectra
for semi-volatile oxygenated organic aerosol (SV-OOA). However, the detailed analysis
exhibited that PMF analysis on smaller dataset sometimes gives mass spectra patterns
identified as LV-OOA and SV-OOA, the latter of which has a high signal at *m/z* 43 and
marginally high signal at *m/z* 44 (*e.g.*, $f_{43} = 0.058$ and $f_{44} = 0.022$ shown in Figure S-15). Thus,
it is worth to note that owing to the large dataset, the overall PMF analysis here may have not
separated a SV-OOA loading from a HOA loading successfully. The time-series variations of
the HOA and LV-OOA mass concentrations showed similar patterns (Figure 3), an implication
that the primary OA and the precursor(s) of LV-OOA are possibly from the same source in large
scale. On average, HOA and LV-OOA accounted for 38% and 59% of the organic aerosols
throughout the study period, respectively. These values are in the same magnitude to the
fractions previously reported during the study in December 2010 (32% and 67% for HOA and
LV-OOA, respectively). In a plot of $f_{43}$ versus $f_{44}$, the data point of $f_{43}$ and $f_{44}$ for LV-OOA (0.043
and 0.237, respectively) in this study was located at the top of the triangle, indicating a high
oxidation state (Figure 4). Meanwhile, the data point of $f_{43}$ and $f_{44}$ for HOA (0.075 and zero,
respectively) was located at the bottom of the triangle, indicating a low oxidation state. The $f_{44}$
/ $f_{43}$ ratios for LV-OOA and HOA were approximately five and zero, respectively. The ratio for
LV-OOA was two times of the high slope in the plot of $m/z$ 44 versus $m/z$ 43 (~2.5) referred in
the previous Sect. (Figure S-14), and the ratio for HOA was lower than the low slope
(approximately one). According to the results of the overall PMF analysis, the observations
shown in Figure S-14 could be explained by a combination of HOA (or SV-OOA) and LV-
OOA. The OM/OC ratio of HOA and LV-OOA were 1.7 and 4.2 $\mu g\ \mu gC^{-1}$, respectively. The
OM/OC ratio of HOA was similar to the ratio of HOA found in the December study, 1.2 $\mu g$
$\mu gC^{-1}$. The OM/OC ratio of LV-OOA was also similar to the ratios of LV-OOA found in our
field studies in December 2010 (3.6 $\mu g\ \mu gC^{-1}$) and March 2012 (4.3 $\mu g\ \mu gC^{-1}$), respectively.
Based on the AMS reference mass spectra available from the web site previously referred,
substances showing such a high OM/OC ratio are only humic-like substances.

**3.5. Chemical clocks**
We used a $NO_x/NO_y$ concentration ratio and a NMHC concentration ratio to explore the extent
of photochemical reaction (*i.e.*, the reaction with OH radical). In this type of chemical clock
analysis, the concentration of a reactive chemical and that of a less reactive chemical are used
in the numerator and the denominator, respectively, of the ratio. As a reaction proceeds, the
numerator decreases while the denominator remains constant; therefore, a change in the ratio
indicates the extent of reaction. In application of chemical clocks to the atmospheric transport
of pollutants, users should be aware of that the extent of reaction may not always be well
defined because emission sources are spatially distributed over the trajectory of an air parcel
in many cases. This type of analysis is ideally suited to situations in which inputs into an air
parcel from additional emission sources during transport are negligible. Our field study for
transboundary air pollution transported over the East China Sea can be the ideal case.

**3.5.1.  $NO_x/NO_y$ clock**
To see if such an assumption is valid, the $NO_x/NO_y$ and hydrocarbon clocks were evaluated.
Given that the conversion of $NO_2$ (the major component of $NO_x$) to $HNO_3$ (one of the
components of $NO_y$)
$$NO_2 + OH \xrightarrow{\ M\ } HNO_3 \qquad\qquad (R1)$$
is the major sink for $NO_x$ and that the concentration of OH radicals, [OH], can be assumed to
be constant, the photochemical age, $t$[OH], of $NO_x$ can be determined according to the
following pseudo-first order rate law:
$$t[OH] = -\frac{1}{k_{NO_2}} \ln \frac{[NO_x]}{[NO_y]}$$ (1)
where [NO$_x$], [NO$_y$], and $k_{NO2}$ are the concentrations of NO$_x$ and NO$_y$ (molecules cm$^{-3}$) at
reaction time $t$ and the temperature-dependent effective second-order rate constant for the
reaction of NO$_x$ with OH radicals, respectively. $k_{NO2}$ includes the concentration of a third body,
[M], which depends on pressure and temperature. To calculate $k_{NO2}$ at ambient temperature and
a pressure of 1 atm, we therefore calculated the third-order rate constant and [M] according to
the method described by Finlayson-Pitts and Pitts (2000) with the polynomial best fit for the
measured ambient temperature mentioned in Sect. 3.1. The calculated $k_{NO2}$ values at 1 atm
ranged from $9.3 \times 10^{-12}$ to $1.1 \times 10^{-11}$ cm$^3$ molecule$^{-1}$ s$^{-1}$, and both the mean and median were
$1.0 \times 10^{-11}$ cm$^3$ molecule$^{-1}$ s$^{-1}$. In turn, the determined $t[OH]$ using the $k_{NO2}$ values and the
[NO$_x$]/[NO$_y$] ratios ranged from $2.9 \times 10^5$ to $1.3 \times 10^8$ (mean ± SD = $(3.4 \pm 1.6) \times 10^7$ h
molecules cm$^{-3}$). We found that the use of a fixed $k_{NO2}$ value (*i.e.*, the mean value of $1.0 \times 10^{-11}$
cm$^3$ molecule$^{-1}$ s$^{-1}$) resulted in biases between –10% and +7% in the estimation of $t[OH]$. We
also found that a temperature variation of ± 5 K resulted in only a ± 5% variation in $t[OH]$.
However, this analysis for the biases does not take into account temperature and pressure
variations during the transport of the air parcels.

The reaction of NO$_2$ with O$_3$, which may result in significant overestimation in the

NO$_x$/NO$_y$ clock, was also evaluated. The reaction of NO$_2$ with O$_3$ forms NO$_3$ radicals:
$$NO_2 + O_3 \longrightarrow NO_3 + O_2$$ (R2).
This reaction channel is important at night, but negligible during the day when $NO_3$ radicals
are quickly photolyzed back to $NO_x$. $NO_3$ radicals react with $NO_2$ to form stable $N_2O_5$, which
is in thermal equilibrium with $NO_2$ and $NO_3$ and therefore acts as a reservoir of $NO_x$:
$$NO_2 + NO_3 \leftrightarrow N_2O_5 \qquad\qquad\qquad (R3).$$
$N_2O_5$ reacts slowly with water to form $HNO_3$, and this process terminates the chain reaction:
$$N_2O_5 + 2H_2O \longrightarrow 2HNO_3 \qquad\qquad (R4).$$
The R4 channel is known to be predominant at the surface of particles (Brown *et al.*, 2006).
Although the nocturnal sink of $NO_x$ by the series of R2-R4 channels may result in a significant
overestimation of $t$[OH], the heterogeneous uptake of $N_2O_5$ (*i.e.*, the R4 channel) is negligible
when sufficient amount of nitrate already exists in particles because the preexisting nitrate
inhibits the forward reaction of R4 channel (Brown *et al.*, 2006). We have observed that there
was enough ammonium that neutralized sulfate and nitrate. Therefore, the negligibly small
heterogeneous uptake of $N_2O_5$ is likely our case. Indeed, a plot of the hourly $O_3$ mixing ratios
(*x*-axis) versus hourly $\ln([NO_x]/[NO_y])$ (*y*-axis) showed no positive correlation, but a clear
inverse correlation ($r^2 = 0.489$), indicating that the turnover of $NO_x$ to $NO_y$ increased as the $O_3$
mixing ratio increased (Figure 5). If the reaction of $NO_2$ with $O_3$ and the subsequent reactions
was the predominant mode of conversion of $NO_x$ to $NO_y$ at night, a positive correlation between
the $O_3$ mixing ratio and the extent of $NO_x$ turnover—that is, between the $O_3$ mixing ratio and
$\ln([NO_x]/[NO_y])$—should be observed in our night time data. Similar observations have been
reported elsewhere (Olszyna *et al.*, 1994; Roussel *et al.*, 1996). Given that during the day, $O_3$
forms only photochemically, this inverse correlation suggests that $NO_x$ conversion was owing
to daytime photochemistry. A conclusion with this possibility was drawn from an analysis of
$O_3$ production efficiency (Yokouchi *et al.*, 2011).

The photochemical reaction of aldehydes is also a sink for $NO_2$, resulting in the

formation of thermally stable peroxyacyl nitrates:
$$RCHO + OH \longrightarrow RCO + H_2O \qquad\qquad (R5)$$
$$RCO + O_2 \longrightarrow RC(O)OO \qquad\qquad (R6)$$
$$RC(O)OO + NO_2 \longrightarrow RC(O)OONO_2 \qquad\qquad (R7)$$
Unfortunately, we cannot evaluate the significance of this channel with our current dataset,
because no data for aldehyde and peroxyacyl radical concentrations are available. Because this
loss channel also occurs in sunlight, the possibility that peroxyacyl nitrate formation
significantly affects $t$[OH] cannot be excluded. The absolute value of $t$[OH] derived from the
$[NO_x]/[NO_y]$ ratio remains uncertain, but as demonstrated by the high correlation between this
ratio and the $O_3$ mixing ratio, the use of the $NO_x/NO_y$ clock nevertheless provides valuable
information about the relative extent of photooxidation. When we plotted the time-series
variation of $t$[OH] estimated from the $[NO_x]/[NO_y]$ ratio (Figure 6), we observed variation
similar to that observed for the hourly average $O_3$ mixing ratio (Figure S-4a), implying a strong
association between the $t$[OH] and the sunlight irradiance.

### 3.5.2. Hydrocarbon clock

When NMHC A and B react with OH radicals at different rate

$$A + OH \longrightarrow Product + H_2O \qquad\qquad (R8)$$

$$B + OH \longrightarrow Product + H_2O \qquad\qquad (R9)$$

$t$[OH] can also be estimated from the ratio of the two NMHCs (Roberts *et al.*, 1984; Rudolph

and Johnen, 1990; Parrish *et al.*, 1992):

$$t[OH] = \frac{1}{(k_A - k_B)} \ln\left( \frac{[A_0]}{[B_0]} \cdot \frac{[B]}{[A]} \right) \qquad\qquad (2)$$

where $[A_0]$ and $[B_0]$ are the initial concentrations (molecules $cm^{-3}$) of NMHCs A and B, which

have short and long lifetimes (relative to each other); [A] and [B] are the concentrations

(molecules $cm^{-3}$) at time $t$; and $k_A$, and $k_B$ are the temperature-dependent rate constants for

reactions of A and B with OH radicals (molecules$^{-1}$ $cm^3$ $s^{-1}$). If NMHCs A and B are emitted

from the same source at the same time, the change in the concentration ratio theoretically

indicates the extent of chemical reaction. However, dilution with an aged air mass containing

depleted NMHCs can also change the NMHC ratio, thus biasing the $t$[OH] estimation (McKeen

and Liu, 1993). This bias can be visualized by plotting two different NMHC ratios with the

same denominator, and we used the [*i*-pentane]/[ethyne] and [toluene]/[ethyne] ratios for this

evaluation. The calculations require the rate constants for the reactions of the NMHCs with OH

radicals at the mean temperature observed, 283.7 K, the mixing ratios of the NMHCs in the

background air, and their initial mixing ratios at emission. Using the Arrhenius equation with
the recommended parameters for *i*-pentane, toluene, and ethyne (NIST Chemistry WebBook,
http://webbook.nist.gov/chemistry/), respectively, the rate constants for the reaction of these
compounds with OH radicals at 283 K (*i.e.*, the mean temperature during the study period)
were calculated to be $3.44 \times 10^{-12}$, $5.88 \times 10^{-12}$, and $7.38 \times 10^{-13}$ cm$^3$ molecule$^{-1}$ s$^{-1}$, respectively.
Note that the variation of the slope for the reaction loss owing to the variation of the
temperature-dependent rate constants between the maximum and minimum temperature (298.3
and 271.5 K) was found to be less than $\pm 2\%$. Thus, the variation of the reactive loss owing to
the temperature change was not influential to our analysis. For the background mixing ratios,
we used mixing ratios observed at Cape Hedo, Okinawa (Kato *et al.*, 2004), which were 0.05,
0.09, and 0.39 ppbv, respectively. For the initial mixing ratios at emission, we used the reported
scores for loadings extracted by means of PMF analysis for the NMHC sources in Beijing
(Wang *et al.*, 2015). The PMF loadings used in the calculations were vehicular emissions 1 and
2, solvent use, and natural gas and gasoline leakage. In addition to these initial mixing ratios,
mixing ratios reported a rural site in northeast China (Lin'an, in the Yangtze River Delta, Tang
*et al.*, 2009) were also tested.

The plot shows that, with respect to the initial NMHC ratio, depletion trends resulting

from use of the solvent-use profile and of the observations in Lin'an deviated substantially
from the observed overall trend (Figure 7). The majority of observed plots lies between the
trends for the dilution with the background air and the reaction loss calculated if the profiles
for the vehicular-emissions and natural-gas and gasoline leakage were used. That is, the
vehicular emissions and the natural gas and gasoline leakage may have been the predominant
emitters of these NMHCs, but source apportionment is difficult because of the uncertainty in
the emission profiles. On the basis of this comparison, we could identify only two possible
significant sources of these NMHCs during the measurement period. The layout of observed
data points in-between the dilution and reactive loss lines also suggests that depletion in their
mixing ratios was a combination of these processes. Comparison of calculated $t$[OH] by the
toluene/ethyne clock with those by the $NO_x/NO_y$ clock exhibited a poor correlation (Figure S-
16), demonstrating the limitation of the toluene/ethyne clock for estimation of $t$[OH] under the
condition at Fukue. A quantitative understanding will require a more sophisticated analysis
based on mass balance with reliable source profiles.

With respect to the background mixing ratios observed at Cape Hedo, the plot also

shows that many of our observed data points were lower than the background NMHC ratios
represented by a brown circle in Figure 7. This result implies that the background NMHC ratios
from the observations at Cape Hedo are still too high to be used as background values of these
NMHC ratios for the study region. It is reasonable to assume that the background mixing ratios
for both toluene and $i$-pentane in the aged air masses were LDL (<3 pptv). This assumption
allows us to approximate the background mixing ratio of ethyne based on the smallest
[toluene]/[ethyne] and [$i$-pentane]/[ethyne] ratios observed. According to the plot, the use of –
3.5 for the ln[toluene]/[ethyne] and –4 for ln[$i$-pentane]/[ethyne], approximately the smallest
ratios observed, seems more reasonable. If we use the highest DL value (3 pptv) as the
background mixing ratio for toluene and $i$-pentane, the background ethyne mixing ratio is then
calculated to be ~0.16 ppbv, which is about 25% of the background value observed at Cape
Hedo by Kato $et\ al.$ (2004). On the basis of the plot in Figure 7, we recommend the use of
0.003, 0.003, and 0.16 ppbv as the background mixing ratios for $i$-pentane, toluene, and ethyne,
respectively, in the region of the East China Sea.

**3.6. Dependence of $f_{44}$ and $O_3$ on $t$[OH]**
A scatter plot of $f_{44}$ as a function of $t$[OH] estimated by the $NO_x/NO_y$ clock showed a
proportional increase of $f_{44}$ with increasing $t$[OH] (estimated by means of the $NO_x/NO_y$ clock)
up to a $t$[OH] value of $7 \times 10^7$ h molecules cm$^{-3}$, and then $f_{44}$ started to level off slightly (Figure
8). That is, $f_{44}$ works as an oxidation indicator below the $t$[OH] of $7 \times 10^7$ h molecules cm$^{-3}$.
The $f_{44}$ oxidation indicator is known to be case dependent, even at this location and below this
upper limit (Irei $et\ al.$, 2015). Considering the existence of HOA during the study period, a
series of findings here and in the previous reports supports our hypothesis that $f_{44}$ varies with
$t$[OH] as LV-OOA, which has a constant and high value of $f_{44}$, mixes with the background-level
HOA, which has a significantly lower constant value of $f_{44}$ than LV-OOA (Irei $et\ al.$, 2014). To
a first approximation of the increasing trend, $f_{44}$ is given by
$$f_{44} = (1.05 \pm 0.03) \times 10^{-9}\, t[\text{OH}] + 0.103 \pm 0.001 \qquad\qquad (3)$$
with an $r^2$ value of 0.369. The first approximation satisfactorily describes the increasing trend
below a $t[\text{OH}]$ value of $7 \times 10^7$ h molecules cm$^{-3}$. The intercept of the first approximation
indicates the $f_{44}$ value for organic aerosol at a photochemical age of zero, that is, $f_{44}$ at emission.
The slope, which was $(1.05 \pm 0.03) \times 10^{-9}$ h$^{-1}$ molecule$^{-1}$ cm$^3$, indicates the rate of increase
of $f_{44}$ as $[\text{OH}]$ is given. Kleinman *et al.* (2007) observed that during the New England Air
Quality Study, the background-corrected $f_{44}$ value increased from 0.08 to 0.13 as –
$\ln([\text{NO}_x]/[\text{NO}_y])$ increased from 0.1 to 1.3, which corresponds to an increase of $t[\text{OH}]$ from 3.2
$\times 10^6$ to $42 \times 10^6$ h molecule cm$^{-3}$. These values give an increase rate of $1.3 \times 10^{-9} \times [\text{OH}]$ h$^{-1}$,
which is almost identical to the rate we calculated in this study. The overall proportionality of
$f_{44}$ with $t[\text{OH}]$ suggests that, like the $\text{NO}_x/\text{NO}_y$ clock, $f_{44}$ worked as an oxidation indicator
during this study period. This, however, is inconsistent with our other report, in which no
proportional increase of $f_{44}$ was observed during the study in different year at the same location
(Irei *et al.*, 2015). Interestingly, our hypothesis of binary mixture of organic aerosol is still
consistent with these contradicting cases.

It has been proposed that the increasing trend of $f_{44}$ can be explained by a binary mixture

of variable amount of LV-OOA depending on extent of reaction processing $x$ for the LV-OOA
precursor and constant amount of HOA (Irei *et al.*, 2014, Supporting Information):
$$f_{44} = \frac{^{HOA}f_{44} \cdot a \cdot (\frac{OM}{OC})_{HOA} + {}^{LVOOA}f_{44} \cdot \left[ 0.3x \cdot b \cdot (\frac{OM}{OC})_{LVOOA} \right]}{a \cdot (\frac{OM}{OC})_{HOA} + \left[ 0.3x \cdot b \cdot (\frac{OM}{OC})_{LVOOA} \right]} \quad (4)$$
In this equation $^{HOA}f_{44}$ and $^{LV\text{-}OOA}f_{44}$ are the fractions of m/z 44 signal for the HOA and LV-
OOA factors from the PMF analysis previously discussed, respectively; (OM/OC)$_{HOA}$ and
(OM/OC)$_{LV\text{-}OOA}$ are the organic mass concentration ratios to the organic carbon concentrations
($\mu g \ \mu gC^{-1}$) for the HOA and LV-OOA from the PMF analysis, respectively; and *a* and *b* values
are arbitrary constants ($\mu gC \ m^{-3}$) that convert the (OM/OC)$_{HOA}$ and (OM/OC)$_{LV\text{-}OOA}$ ratios to
the organic mass concentrations of the HOA and the LV-OOA, respectively. The factor "0.3",
which is multiplied by the variable *x*, is a factor for the SOA carbon yield based on the
laboratory experiments of SOA formation by toluene photooxidation (Irei *et al.*, 2006; Irei *et*
*al.*, 2011). The equation (4) has one variable, *x*, and 6 parameters, four of which are determined
by PMF analysis. The greater extent of reaction proceeds, the greater LV-OOA contributes to
the binary mixture of HOA and LV-OOA, each of which has significantly different $f_{44}$ value.
Consequently, the $f_{44}$ of the binary mixture containing a significantly low $f_{44}$ continues to
increase until it is saturated with LV-OOA. This hypothesis consistently explains our
observations that the $f_{44}$ oxidation indicator sometimes worked, and sometimes did not. There
is also a possible limitation that the indicator also depends on distinctive values of $f_{44}$. If two
members had the similar $f_{44}$ values, the indicator would not work.

The $f_{44}$ curve of organic aerosols was calculated using three different combinations of

parameters listed in Table 3. It was found that the model calculation underestimated the $f_{44}$
(Figure 8) when 0.05 and 1 $\mu$gC m$^{-3}$ were used for the $a$ and $b$ values, respectively, together
with the rest of the parameters obtained from the PMF analysis (*i.e.*, applying the parameters
in the combination I in Table 3). Although these $a$ and $b$ values were used in the previous report
and demonstrated reasonable agreement with the observations (*i.e.*, applying the parameters in
the combination III), the agreement was owing to different $f_{44}$ values and OM/OC ratios
extracted from the PMF analysis (see the Section 3.4). To have reasonable agreement with the
observations using the $f_{44}$ and OM/OC extracted by the PMF analysis in Sect. 3.4, the use of
0.025 and 1 $\mu$gC m$^{-3}$ for the $a$ and $b$ values (applying the parameters in the combination II) was
found to give the best fitting to the observations.

As discussed previously, there was a strong relationship between the NO$_x$ turn over and O$_3$

mixing ratio (Sect. 3.5.1). This relationship can be converted to the one between $t$[OH] and O$_3$
mixing ratios (Figure 9). An obtained linear relationship was [O$_3$] = (3.48 $\pm$ 0.06) $\times$ 10$^{-7}$ $\times$
$t$[OH] + 30.7 $\pm$ 0.3. This provides the increasing rate of ozone $(3.48 \pm 0.06) \times 10^{-7} \times$ [OH] ppbv
h$^{-1}$ and the background ozone mixing ratio of 30.7 ppbv in this region. If [OH] of 5$\times$10$^5$
molecules cm$^{-3}$ (Takegawa *et al.*, 2007; Irei *et al.*, 2014) is given as the mean concentration of
OH radical during the long-range transport in this region, the equation gives the average ozone
production rate of 0.174 ppbv h$^{-1}$. A combination with measurements for OH radical
concentration will secure a more accurate production rate of ozone in this region.
**4.    Summary**
To improve our understanding of the ozone and SOA formation from the oxidation of
atmospheric pollutants, we conducted field studies from December 2010 to May 2011 on Fukue
Island, Nagasaki Prefecture, Japan. Wind-sector analysis of CO mixing ratios revealed that the
ratio showed almost no wind-sector dependence, suggesting that the influence of emissions
from residential areas near the measurement site was negligible. This fact in turn indicates that
the influence of mid- and/or long-range transport of air pollutants to the site had a significant
influence. Photochemical age, $t$[OH], was estimated from [$NO_x$]/[$NO_y$] and a NMHC
concentration ratio, and the validity of the ratios was evaluated. The evaluation suggested that
the hydrocarbon clock was significantly influenced by mixing with background air containing
0.16 ppbv of ethyne, a NMHC with a relatively long lifetime, resulting in significant bias in
the estimation of $t$[OH]. In contrast, loss of $NO_x$ owing to reaction with $O_3$ at night was not
influential to the $NO_x$/$NO_y$ clock, which thus seemed to function appropriately, at least with
respect to relative aging. The $t$[OH] value obtained with the $NO_x$/$NO_y$ clock was then compared
with $f_{44}$ obtained by AMS measurements, and $f_{44}$ was observed to increase with increasing
$t$[OH], indicating the $f_{44}$ can also be used as an oxidation indicator. This indicator likely works
under the condition where two different types of organic aerosols, such as primary and
secondary organic aerosols represented by hydrocarbon-like organic aerosols and low-volatile
oxygenated organic aerosol, respectively, are mixed. Using linear regression analysis, we
estimated that the $f_{44}$ increase rate for organic aerosols transported over the East China Sea
averaged $(1.05 \pm 0.03) \times 10^{-9} \times$ [OH] h$^{-1}$. This rate was almost identical to the background-
corrected rate of $1.3 \times 10^{-9} \times$[OH] h$^{-1}$ observed during the New England Air Quality Study in
the summer of 2002. The consistency may imply the production of similar SOA component(s),
possibly humic-like substances. In addition, a clear proportional relationship was observed
between $O_3$ and $t$[OH]. According to the linear regression analysis, the increase rate and
background mixing ratio of $O_3$ in this region were found to be $(3.48 \pm 0.06) \times 10^{-7} \times$ [OH] ppbv
h$^{-1}$ and 30.7 ppbv, respectively.
**Author Contribution**
Satoshi Irei contributed to the AMS, $O_3$, and meteorological measurements and is the person
in charge of the data analysis and writing the manuscript. Akinori Takami is the person in
charge of the AMS, $O_3$, and meteorological measurements. Yasuhiro Sadanaga is the person in
charge of the $NO_x$ and $NO_y$ measurements. Seiichiro Yonemura is the person in charge of the
CO measurements. Yoko Yokouchi is the person in charge of the NMHC measurements.
Susumu Nozoe contributed to the NMHC measurements. Hiroshi Bandow contributed to the
$NO_x$ and $NO_y$ measurements.
**Acknowledgements** We acknowledge the NOAA Air Resources Laboratory (ARL) for the
provision of the HYSPLIT transport and dispersion model and/or READY website
(http://www.ready.noaa.gov).This project was financially supported by the Special Research
Program from the National Institute for Environmental Studies, Japan (SR-95-2011). The
project was partially supported by the International Research Hub Project for Climate Change
and Coral Reef/Island Dynamics of University of the Ryukyus and the ESPEC Foundation for
Global Environment Research and Technologies (Charitable Trust).

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

Figure captions

Figure 1. Location of the Fukue Island monitoring station.

Figure 2. Extracted mass spectra from two-factorial PMF analysis: top, mass spectra identified
as hydrocarbon-like organic aerosol (HOA); bottom, mass spectra identified as low-volatile
oxygenated organic aerosol (LV-OOA).

Figure 3. Temporal variation of mass concentration of HOA (orange) and LV-OOA (green)
obtained by PMF analysis.

Figure 4. Plot of $f_{44}$ versus $f_{43}$ for different types of organic aerosols extracted from PMF
analysis. Dashed lines are the limits of oxidation states reported by Ng $et\ al.$ (2010).

Figure 5. Scatter plot of natural logarithm of $[NO_x]/[NO_y]$ ratio versus $O_3$ mixing ratio. The
data points with the ozone mixing ratios less than 25 ppbv were excluded from the linear
regression.

Figure 6. Time-series variation of photochemical age, $t$[OH], estimated from $[NO_x]/[NO_y]$
ratios.

Figure 7. Scatter plot of natural logarithm of [toluene]/[ethyne] ratio as a function of natural
logarithm of [$i$-pentane]/[ethyne] ratio (gray dots). Linear regressions shown are calculated
depletion trends resulting from mixing with background air (dotted lines) and from reaction
with OH radicals (solid lines); these trends were determined by using the initial NMHC ratios
from the literature, for vehicular emissions 1 (black open circle), vehicular emissions 2 (red
open circle), solvent use (green open circle), and natural gas and gasoline leakage (blue open
circle) observed in Beijing (Wang $et\ al.$, 2015), as well as field measurement data obtained at
Lin'an, a rural background site in the Yangtze River Delta, China (yellow open circle) from
Tang $et\ al.$ (2009). The brown open circle that all the dotted lines meet at corresponds to the
background values observed at Cape Hedo (Kato *et al.*, 2004). See the text for the calculation
and references for these data.

Figure 8. Scatter plot of hourly averaged $f_{44}$ (black dot) as a function of photochemical age,
$t$[OH], estimated by means of the $NO_x/NO_y$ clock (the bottom *x*-axis) and a linear regression
(grey line). As comparison, $f_{44}$ binary mixing models (dotted curves) of HOA and LV-OOA
using different combinations of model parameters (combination I (green); combination II
(blue); and combination III (red)) are also shown. See the text for the detail of the combinations
of model parameters.

Figure 9. Scatter plot of ozone mixing ratio versus photochemical age ($t$[OH]). The data
points with the ozone mixing ratios less than 25 ppbv were excluded from the linear
regression.












**Table 1. Concentrations and mixing ratios of chemical species observed during the study period.**

| | Number of data | Mean | SD | Min[a] | Max | Lower quartile | Median | Upper quartile |
|---|---|---|---|---|---|---|---|---|
| Fine PM | | | | | ($\mu$g m$^{-3}$) | | | |
| Chloride | | 0.08 | 0.12 | LDL | 2.65 | 0.03 | 0.04 | 0.09 |
| Ammonium | | 1.5 | 1.6 | LDL | 14.7 | 0.6 | 1.1 | 1.8 |
| Nitrate | | 0.69 | 1.43 | LDL | 22.00 | 0.12 | 0.25 | 0.63 |
| Sulfate | | 4.2 | 3.3 | LDL | 23.8 | 2.0 | 3.4 | 5.5 |
| Organics | 22726 | 2.7 | 1.9 | LDL | 24.5 | 1.4 | 2.2 | 3.4 |
| Total[b] | | 9.2 | 7.4 | 0.02 | 66.7 | 4.7 | 7.5 | 11.2 |
| $m/z$ 43 in organics | | 0.18 | 0.14 | LDL | 4.17 | 0.08 | 0.14 | 0.22 |
| $m/z$ 44 in organics | | 0.40 | 0.30 | 0.06 | 2.45 | 0.20 | 0.33 | 0.50 |
| $m/z$ 57 in organics | | 0.03 | 0.04 | 0.01 | 1.87 | 0.02 | 0.03 | 0.04 |
| | | | | | | | | |
| Gas-phase species | | | | | (ppbv) | | | |
| CO | 4163 | 230 | 102 | 57 | 1136 | 160 | 204 | 272 |
| NO | 4176 | 0.06 | 0.16 | LDL | 4.45 | LDL | LDL | 0.06 |
| NO$_x$ | 4176 | 1.39 | 1.16 | LDL | 12.70 | 0.70 | 1.10 | 1.70 |
| NO$_y$ | 4163 | 4.86 | 3.49 | 0.13 | 25.41 | 2.49 | 3.95 | 6.03 |
| O$_3$ | 4165 | 45 | 11 | 10 | 97 | 38 | 45 | 52 |
| $i$-Pentane | 3856 | 0.106 | 0.079 | LDL | 2.055 | 0.066 | 0.098 | 0.132 |
| Toluene | 3856 | 0.110 | 0.142 | LDL | 2.625 | 0.044 | 0.071 | 0.123 |
| Ethyne | 3856 | 0.496 | 0.326 | 0.014 | 4.442 | 0.304 | 0.407 | 0.597 |

[a]LDL: lower than detection limit.

[b]Sum of chloride, ammonium, nitrate, sulfate, and organics.

**Table 2. Coefficients of determination for correlations between chemical species concentrations.**

|  | PM_NH$_4$ | PM_NO$_3$ | PM_SO$_4$ | PM_org | $m/z$ 43 | $m/z$ 44 | $m/z$ 57 | O$_3$ | NO$_x$ | NO$_y$ | CO | $i$-Pentane | Toluene | Ethyne |
|---|---|---|---|---|---|---|---|---|---|---|---|---|---|---|
| PM_NH$_4$ | 1 | 0.693 | 0.639 | 0.696 | 0.443 | 0.755 | 0.323 | 0.251 | 0.007 | 0.480 | 0.405 | 0.004 | 0.026 | 0.097 |
| PM_NO$_3$ | 0.693 | 1 | 0.263 | 0.529 | 0.389 | 0.521 | 0.320 | 0.145 | 0.035 | 0.544 | 0.314 | 0.025 | 0.051 | 0.107 |
| PM_SO$_4$ | 0.639 | 0.263 | 1 | 0.430 | 0.380 | 0.463 | 0.191 | 0.128 | 0.001 | 0.179 | 0.371 | 0.013 | 0.006 | 0.125 |
| PM_org | 0.696 | 0.529 | 0.430 | 1 | 0.747 | 0.949 | 0.606 | 0.303 | 0.053 | 0.559 | 0.562 | 0.060 | 0.081 | 0.198 |
| $m/z$ 43 | 0.443 | 0.389 | 0.380 | 0.747 | 1 | 0.640 | 0.588 | 0.146 | 0.153 | 0.459 | 0.543 | 0.100 | 0.094 | 0.301 |
| $m/z$ 44 | 0.755 | 0.521 | 0.463 | 0.949 | 0.640 | 1 | 0.471 | 0.384 | 0.016 | 0.510 | 0.526 | 0.007 | 0.039 | 0.142 |
| $m/z$ 57 | 0.323 | 0.320 | 0.191 | 0.606 | 0.588 | 0.471 | 1 | 0.098 | 0.160 | 0.417 | 0.394 | 0.106 | 0.137 | 0.236 |
| O$_3$ | 0.251 | 0.145 | 0.128 | 0.303 | 0.146 | 0.384 | 0.098 | 1 | 0.007 | 0.292 | 0.288 | 0.013 | 0.006 | 0.053 |
| NO$_x$ | 0.007 | 0.035 | 0.001 | 0.053 | 0.153 | 0.016 | 0.160 | 0.007 | 1 | 0.309 | 0.136 | 0.195 | 0.225 | 0.221 |
| NO$_y$ | 0.480 | 0.544 | 0.179 | 0.559 | 0.459 | 0.510 | 0.417 | 0.292 | 0.309 | 1 | 0.674 | 0.117 | 0.155 | 0.422 |
| CO | 0.405 | 0.314 | 0.371 | 0.562 | 0.543 | 0.526 | 0.394 | 0.288 | 0.136 | 0.674 | 1 | 0.193 | 0.126 | 0.724 |
| $i$-Pentane | 0.004 | 0.025 | 0.013 | 0.060 | 0.100 | 0.007 | 0.106 | 0.013 | 0.195 | 0.117 | 0.193 | 1 | 0.410 | 0.435 |
| Toluene | 0.026 | 0.051 | 0.006 | 0.081 | 0.094 | 0.039 | 0.137 | 0.006 | 0.225 | 0.155 | 0.126 | 0.410 | 1 | 0.302 |
| Ethyne | 0.097 | 0.107 | 0.125 | 0.198 | 0.301 | 0.142 | 0.236 | 0.053 | 0.221 | 0.422 | 0.724 | 0.435 | 0.302 | 1 |

**Table 3. Three different combinations of model parameters.**

| Parameters | Combination I[a] | Combination II[a] | Combination III[b] |
|---|---|---|---|
| a ($\mu$gC m$^{-3}$) | 0.05 | 0.025 | 0.05 |
| b ($\mu$gC m$^{-3}$) | 1 | 1 | 1 |
| $^{HOA}f_{44}$ | 0 | 0 | 0.08 |
| $^{LV\text{-}OOA}f_{44}$ | 0.237 | 0.237 | 0.22 |
| (OM/OC)$_{HOA}$ ($\mu$g $\mu$gC$^{-1}$) | 1.7 | 1.7 | 1.2 |
| (OM/OC)$_{LV\text{-}OOA}$ ($\mu$g $\mu$gC$^{-1}$) | 4.2 | 4.2 | 3.7 |

[a]The $f_{44}$ and OM/OC values for HOA and LV-OOA are based on the results from the PMF analysis.

[b]Parameters used in the previous report (Irei *et al.*, 2014, supportive information).