# Peer review of "Photochemical age of air pollutants, ozone, and secondary organic aerosol in"

_Atmospheric Chemistry and Physics, 2015_

## Referee Comment (RC1) · Anonymous Referee #2 · 10 Feb 2016

General Comments: My understanding of the purpose of this paper was to determine the age of air masses based on gas and particle phase oxidation products transported downwind of a highly polluted region. This work is motivated by a desire to understand the strong and evolving influence that point sources of air pollution in China have on the surrounding regions. It is clear in this paper that a lot of experimental work was done. The analyses performed are explained well and are thorough. All of the results are presented well, but I feel conclusions drawn from the results could be improved. For instance, what does it mean to have a similar f44 increase rate at your site as was observed during the New England Air Quality Study? I'm having trouble bridging the conceptual gap with how determining the age of these transported air masses

combined with the chemical oxidation product information gives us useful information about SOA or transported air pollution generally. Having this explicitly and simply laid out would be very useful, but it is not necessary.

Specific and technical comments:

section 3.3 – line 15 or so... I would change "highly correlated" to just "correlated"... that might be even stretching it with an R2 of 0.562 for OA but with ambient measurements "correlated" seems fair

section 3.3 – line 21 or so ... can you site a source for your average OH concentrations? generally to equate OH concentration into an equivalent "OH exposure day" in chamber studies we will use 1.5x10ˆ6 [Mao, et al. 2008].

section 3.3 – "The high correlation (of particulate ammonium) with [delete organics] organics (m/z 44) suggests that (organics are primarily composed of carboxylic acids). [delete beyond here] in major the organics composed of carboxylic acids.

section 3.3 – I didn't really notice before here, but ammonium (NH4+, which is measured by the AMS is referred to as "ammonia" here which is not correct)

section 3.4 – what are the correlation values between your extracted spectra from PMF and the spectral database? What are the correlation values between your December spectra and the spectra described in this study?

General comment – The term "signifigance" is used, generally, in scientific literature to describe a statistical significance. When "the significance of a reaction channel" is being evaluated you could alternitavely say "the relevance". You could also say something like "the reaction of x with y is dominant during the day" as opposed to significant.

Page 21 - grammatical correction "This hypothesis consistently *explains our observations that the f44 oxidation indicator sometimes worked, and sometimes did not."

---

## Referee Comment (RC2) · Anonymous Referee #3 · 13 Feb 2016

Overview: This manuscript presents measurements of mixing ratios of selected trace gases and composition of sub-micron secondary organic aerosol at a rural location in Southern Japan. The authors compare these measurements to previous measurements from this site and other sites in the region, discuss correlations between measured species, and estimate the photochemical age of air masses using concentration ratios of selected trace gases. The stated purpose was to use these age estimates to explain variation in the concentration of secondary pollutants, specifically SOA and ozone. Understanding the evolution of SOA and factors influencing ozone formation are highly relevant areas of research especially in the southeast Asia region. The major strength of this manuscript is the high quality dataset generated at a site that receives

air masses from industrialized areas of China after a short transit across the East China Sea. The experimental approach is straightforward, but sound, and the methods used are adequately described. The paper is, for the most part, well written with clear organization. Ultimately, the authors present an independent, quantitative estimate of the relationship between photochemical age and oxidized organic content of SOA; however, the manuscript does not clearly address the stated objective of determining the relationship between photochemical age and ozone mixing ratios. While this relationship is discussed in lines 362-368 and 385-392 and in Figure 10, the manuscript could benefit from a clearly identified and consolidated discussion of this relationship similar to that in section 3.6 for f44 vs. t[OH]. Alternatively, the mention of ozone in line 79 should be eliminated. In addition, several other issues should be addressed to improve the clarity of the manuscript as detailed below.

Specific comments:

Title: The term "oxidation products" could be more specific as many readers would consider this to imply that gas phase oxidation products (OVOCs) were measured. Perhaps mention SOA or use the term "secondary air pollutants".

Abstract: It may be beneficial to mention the range of t[OH] calculated using NOx/NOy. Given the stated purpose of the study, some mention of the relationship between ozone and NOx/NOy should be included. Can the importance of the calculated f44 increase rate be put in better context in final sentence instead of a simple comparison to the NEAQS data?

Introduction: The sentence on lines 50-51 indicates that air masses move from east to west; this is opposite of the direction from China to Japan. Otherwise, this section is clear and concise.

Experimental: This section is also sufficiently thorough, but also concise. A more detailed description of potential local sources of trace gases in the study area would be beneficial for readers unfamiliar with Fukue Island. For example, are there agricultural operations that could contribute to the high particulate nitrate concentrations, or combustion sources other than automotive traffic that could contribute to ethyne and CO?

Results and discussion: Section 3.1: Lines 139-140: How were precipitation events determined to have a negligible effect on trace gas and aerosol data?

Section 3.2: Lines 167-173: Are there agricultural operations in this region that could contribute to the high particulate nitrate concentrations? Lines 224-228: The seven high-concentration episodes were derived from industrialized areas of the Asian continent. This text should be added because the clean air masses originating in Mongolia are transported from the Asian continent, too.

Section 3.3 Lines 242-243: Ethyne is a tracer for combustion sources in general, not just vehicular emissions. Line 250: Explain what variables were used in the regression. Was it CO and NOy? Lines 253-254: What do you mean by recently improved emission of NOy? Reduced emissions?

Section 3.4 Lines 289-290: What was the match percentage for your HOA and LV-OOA spectra compared to the Ulbrich database? Lines 299-300: What does the high OM/OC ratios similar to humic-like substances say about the sources of your observed OA?

Section 3.5: Temperatures in this section are given in K and °C. Choose one unit and be consistent throughout the manuscript. Also, at some point in this section, it is important to explicitly state that no reliable t[OH] was calculated from the hydrocarbon clock. Lines 313-315: Consider rephrasing these sentences to indicate that because there are few potential sources of these gases between emission sources in Asia and the study site, this study offers an opportunity to use photochemical clock estimates under nearly ideal circumstances. Lines 374-376: The difference is consistent with what? Please clarify your meaning. Lines 446-447: These values refer to "natural log-transformed hydrocarbon ratios" Line 498: Significantly low what? It looks like

something is missing in this sentence.

Section 3.6: This discussion could benefit from a clear statement comparing the proportions of HOA and LV-OOA observed in the 2014 measurements (Irei et al. 2015) and in the current study and the correlation coefficient with t[OH] or NOx/NOy for each dataset. This may be helpful in determining a minimum proportion of LV-OOA necessary to use f44 as an indicator of oxidation.

Summary: Again, the use of the term "oxidation products" could be more specifically referred to as oxidized organic particulate matter, and some mention of the relationship between ozone and t[OH] should be made.

Figures: Figures 2 and 3: Can these be combined to include wind direction in Figure 1? If not, it would be helpful to have percentages on the wind rose in Figure 3 to indicate the distribution of wind direction observations. Figure 5: This figure is quite large. Can panels with species with similar concentration ranges be combined? Also in panel (e), why does the baseline concentration of isopentane decrease after the break in the data? Was this a calibration issue? Figure 6: A boxplot of CO mixing ratios binned by wind direction may be more useful in demonstrating the lack of wind dependence. Or adding a mean or median line to the wind rose would help. Figure 12: It would be helpful to show an overall trendline for the data to allow for a comparison with the modeled trends.

Tables: Double check that consistent significant figures are used in all tables. In Table 1, for example, the Max CO mixing ratio is given to 4 significant figures, the median to 3 sig figs and the minimum to 2 sig figs.

---

## Referee Comment (RC3) · Anonymous Referee #4 · 15 Feb 2016

Review of "Photochemical age of pollutants and oxidation products in transboundary air observed on Fukue Island, Nagasaki, Japan" for Atmospheric Chemistry and Physics

The authors have collected an interesting data set of trace gas and aerosol observations from a site in Japan which is exposed to continental outflow from the Chinese mainland. The title leads with "Photochemical age". Figure 10 based on NOx/NOy shows a reasonable trend in that there is more ozone in older air masses. There is a link between photochemical age and f44, though very noisy. An apparent conflict with the authors earlier work is examined with a model that gives f44 in terms of the properties of HOA and LVOOA, the amounts and properties of which differed between campaigns. A parameterization is arrived at with multiple constants for fitting, some

of which may be derivable. That aspect deserves discussion. Unfortunately the differences between campaigns is not fully resolved.

In regard to the trajectory analysis I recognize that the accuracy of individual trajectories is generally not high enough to make definitive statements. When considered in groups one can gain insights as to source information. I believe that the source identification would be more persuasive if the experimental period were divided into sets with 1) episode levels of CO and 2) mid or low levels of CO and the ensemble of trajectories for these conditions compared.

In regard to photochemical age: There are many ways in which ratios can give biased age. In parts of this paper photochemical age is treated as having quantitative potential, as in the discussion of rate constant for OH+NO2. But in the end the authors seem to get it right, a valuable tools to give information on the relative effects of atmospheric processing

Some of my comments are ways in which this manuscript could be improved. Some are critical. There are conclusions, which may or may not be correct, but are not substantiated. There are serious errors concerning nighttime chemistry of NOy and the lifetime of NOy which need to be corrected before this manuscript is considered for publication.

Comments

Abstract, line 9-11 correlation of NOy with CO implies negligible loss of NOy by reaction with OH radical. Loss from the NOy family of compounds due to reaction with OH is very slow. Loss of NOy is mainly from wet and dry deposition. Oxidation of NO2 to form HNO3 is rapid, but does not lead to loss of NOy. HNO3 so formed could partition to the aerosol phase but still would be detected as NOy by a NOx dectertor with a heated Mo catalyst.

Abstract, line 21. the f44 increase rate was 1.05(-9) hˆ(-1) moleculeˆ(-1) cmˆ(-3). The

terminology is confusing. f44 is dimensionless and its rate of change should have units of 1/time. The f44 increase rate is actually given by 1.05(-9) hˆ(-1) moleculeˆ(-1) cmˆ(-3) [OH], where [OH] has units of molecules/cmˆ3. The presentation in Eq. 3 is correct.

Page 5 Discussion of inlets and NOy measurement. Given the inlet dimensions and flow rate, there is approximately a 30 second transit time from the top of the inlet to the NOy instrument. Unlikely that HNO3 would make it through. A description of the measurement used in the present study should include the converter location

Page 11, line 15-17 CO was highly correlated with NOy (rˆ2 = 0.674), ethyne (rˆ2 = 0.724) and organic aerosols (rˆ2 = 0.562) These correlations are not that high in comparison to observations in other studies. I do not know if the modifier "highly" is warranted. The present data set is accumulated over a few seasons and at a location that has long range transport from multiple directions. The diversity of emission sources and degrees of atmospheric processing will suppress the overall correlation.

Page 11, lines 20-21 at an average OH of 5e(5) the lifetime of NOy is 1.7 days. This is incorrect. 1.7 days is the right order of magnitude for oxidation of NO2 by OH under the assumed low OH concentration. Oxidation of NO2 transforms one member of the NOy family to another. It does not change the NOy concentration. The lifetime of NOy is set by wet and dry deposition and is affected by OH only in so far as reaction changes wet and dry deposition rates.

Page 11, line 23-25 NOy and CO were transported over a short distance This conclusion relies on a 1.7 day lifetime for NOy and is therefore suspect.

Page 11 – 12. Discussion of NOy to CO ratio. The observed ratio from a linear regression is 0.03. A study published in 2002 gave a ratio of 0.1 and a model calculation published in 2012, for air masses which had been transported long distances, gave a ratio of 0.03. The change in the ratio from the 2002 study to that from the present paper and the 2012 study is ascribed to recently improved emission of NOy.

The text implies that these few scattered numbers are due to emission changes. If that claim is made it should be documented. In the U.S., emission controls have caused urban ratios of NOx to CO to increase, at least up to around 2010 (see work by Parrish et al). Low NOy/CO ratios of order 0.03 are usually indicative of long range transport in which NOy has been partially removed by deposition. A minor point: One usually does not refer to NOy emission but rather NOx emissions.

Page 11, line 4-5 and following. high correlation of particulate organics and ammonium suggest that in major the organics composed of carboxylic acids In my opinion this conclusion, true or not, is not demonstrated by the data provided. An air mass that comes from a polluted region is likely to have high concentrations of multiple pollutants. Thus a correlation between organics and NH4 could be due to 1) a correlation between sulfate and organics and 2) a correlation between sulfate and ammonium. It is difficult to disentangle multiple interactions. One could start with multi-variable regressions or PMF calculations that include inorganic ions.

Page 15, line 7 to Page 16, line 6 Paragraph on NO3 chemistry. The gas phase reaction of N2O5 with H2O is very slow. Removal of N2O5 by reaction with water occurs in aerosol. It is rapid for acidic aerosol but slow for near-neutral aerosol (Brown et al, Science, 311, 67-70, 6 Jan 2006; Zaveri et al, JGR 115, D12304, 2010). If the N2O5 is not removed, NOx will be regenerated. NO3 can also react with VOCs, primarily olefins. Depending on conditions, nighttime chemistry can remove most or almost no NOx. The reaction rate cited for NO2 + O3 is just the one way flux through one of a series of reactions.

Page 16, line 7-20 Relative effects of nighttime and daytime NOx chemistry on O3. The effects of daytime chemistry have to be considered. Photochemical O3 production in the day is a chain reaction creating several O3 for each NOx removed. By taking differences between the 75th and 25th percentile data in Table 1 (best I could do with data on hand) one obtains a qualitative estimate of 7.5 molecules of Ox produced per molecule of NOx oxidized. The actual value is lower by an unknown amount because

of NOy deposition. Nighttime chemistry is less efficient in using NOx to remove ozone. If the same amount of NOx is removed in the day and night, there will still be a positive correlation between O3 increase and NOx decrease.

Page 19, Section 3.6 Dependence of f44 on t[OH] There is some increase in f44 with age. Perhaps this would show up better if the data was binned or lowess fitted. It is puzzling that different studies gave different results. Eq. 4 makes sense. I don't understand why a and b are arbitrary parameters. What must be measured to get their values?

Minor points

Page 5, Line 23-24. Could you please supply DL for NO and NO2. I am surprised to see a single figure for both as the measurement of NO2 is done by subtraction and involves the LED efficiency.

Page 6, line 16-17 Are the AMS detection limits for a 10 minute period?

Page 6, line 23 and 25 What averaging times are used in specifying detection limits for NOx, NOy, O3, and CO?

Page 14-15 Photochemical age, in particular choice of reaction rate constant for OH+NO2. The range of values due to temperature and pressure is small compared with other systematic errors such as the ratio of NO2 to NOx and the occurrence of reaction channels (e.g. PAN formation) that remove NO2.

Page 15, line 1-5 I am confused by the sensitivity calculation. As I understand, kNO2 is between 9.3e-12 and 1.1e-11. A nominal value of 1.0e-11 was used in the calculations. The product of kNO2 *t[OH] must remain constant as it is determined by a measured ratio of NOx to NOy. From Eq. (1) if 1.1e-11 is used in place of 1.0e-11, t[OH] decreases by 9%. Going the other way, t[OH] increases by 8%. The stated range in bias is different; -10% and 5%. A change in temperature of plus or minus 5K is brought up but evidently is not what is used in arriving at the kNO2 values in line 330. But more

important why bother with the extended discussion of the temperature dependence of NO2+OH, when there are much more significant factors. Factors left out are the ratio of NO2 to NOx and the occurrence of other reaction such as PAN formation.

Page 20, line 19 extent of reaction Needs a definition.

Figure 9. Why is the f44 of HOA exactly zero? OA/OC for this PMF component has an O to C ratio approximately equal to one.

Supplement The text implies that Figures S1 to S-7 show trajectories for end of each episode terminating in Pacific Ocean or Mongolia. This is a hard feature to pick out. In some cases (S-7) the last trajectory passes over the same regions as trajectories that are part of the episode, but with a greater wind speed. In Fig S2 the last trajectory terminates over the East China Sea. However, this trajectory is shorter than the others and appears to point toward the mainland.

Table S1 The main text, line 223, promises qualitative information on the concentrations of other (non-CO) chemical species. I was expecting average or peak values, not check marks. The foot note to Table S1 specifies that the check marks are for observation of remarkably high concentrations without specifying what "remarkably high" means, either on an absolute basis or relative to the average or frequency distribution of the ensemble of measurements. Additional information needs to be added to Table S1.

TYPOs, wording Page 4, line 2 "east to west" Should be west to east

Page 9, line 1. emission sources of nitrate Should be emission sources of NOx.

Page 11, line 5 in major the organics composed of carboxylic acids Suggest: organics are primarily composed of carboxylic acids

Page 21, line 7 photoxidation toluene Suggest: toluene photoxidation

Page 21, line 8 ".. parameter, the 4 of which are determined by PMF analysis" Eliminate "the"

Page 21, line 8 starting with "More progress.." Not a sentence.

Page 21, line 10 "containing a significantly low continues to increase. Words are missing

Page 39, Symbols on Figure 6. Colors for top two categories difficult to distinguish. I have normal color vision.

Page 41, x-axis of Figure 8. Dates should be the same as used in other figures, i.e. Dec 1, Jan 1, Feb 1, etc.

Figure 11 appears to be missing from last version that I downloaded. In a previous version it had a time axis that did not match others figures.
* * *

---

## Referee Comment (RC4) · Anonymous Referee #1 · 16 Feb 2016

The paper focuses on whether the use of the fractional contribution of the m/z 44 signal to the total organic aerosol mass spectra (f44) can be used as an oxidation indicator. The authors have collected a certain amount of measurements of trace gases in order to draw their conclusions and make an educated guess for the missing information. The factor f44 was compared with the photochemical age of the pollutants (t[OH]) calculated from NOx/NOy, as the toluene/ethyne concentration ratio (NOx/NOy and hydrocarbon clock, respectively) is not applicable in this situation and it was found to increase as t[OH] increased. This led to the conclusion that the factor f44 can be used as oxidation indicator and a discussion about the applicability of this method and possible causes of discrepancy with previous studies was included.

[Figure]

The manuscript is in generally clear though I think some improvements in the quality of the figures are necessary. I would suggest considering moving most of the time series of trace gases to the SI or at least to create a more compact one page figure with all of them together. As they are now they do not really improve the quality of the paper or give such additional information that requires them to be plotted individually.

One major point I have is the choice of the average OH concentration for the lifetime of CO, ethyne and NOy in section 3.3 and for the estimation of the competition of the O3 reaction in section 3.5. It is not clear how the authors arrive at the estimated concentration of average OH. The authors should clearly justify their choice, and as it is an estimate they should give a good idea of the effect of changing the estimation, i.e. the sensitivity of their estimate. Later in that section (lines 17, page16) it is reported that a value of 3 x 106 molecules cm-3 would be 6 times large than the concentration reported previously by Irei et al. 2014. As far as I can tell, there is no measured OH concentration reported in the cited paper.

In addition I think that figure 9 requires some additional explanation as it is hard to understand and as some suggestions for values of background mixing ratios of NMHC in the region of East China Sea are drawn from this figure. I would suggest including a legend to make it easier to immediately identify which color is which. Would it be possible to highlight the observed trend that on line 17 on page 18 is lying between the trends for the dilution with the background air and the reaction loss? Also, how can I see (line 10 on page 19) from figure 9 that the background NMHC ratios seem to lie between -3.5 and -4?

Specific comments:

In general I would suggest to remove the "Note that" from the manuscript.

- Line 2, page 4: the wind is from west to east

- Lines 25 to 27 page 4: it is not totally clear what is given the evidence of SOA

[Figure]

production

- Figure S2: I would suggest putting the cardinal points directly on the figure

- Table 1: what does n stand for?

- Lines 10 to 12 page 9: It is reported that the concentrations of organic aerosol reported by Irei at al. 2014 were relatively low. How does this fit with the current study?

- Lines 10 to 13, page 11: It is not clear to me the meaning of this sentence. High mixing ratios occurred for a small portion of the observed data or high mixing influenced by local pollution occurred for a small portion of the observed data. Is it possible to distinguish between the two cases?

- Line 1and 2 page 12: what is the meaning of recently improved emissions in $NO_y$?

- Line 13, page 13: which data point?

- Line 28, page 13 to line 2, page 14: I suggest rephrasing the sentence.

- Line 3 page 16: 1 x 106 molecules cm-3 of OH radicals do not correspond to 0.05 ppbv…rather to 5 x10-5 ppbv.

- I would suggest to separate in section 3.6 the two different t[OH] estimates with two subsections. It should also be stated in a clearer way that the estimate of t[OH] from the hydrocarbon clock is not possible for this conditions.

---

## Author Comment (AC1) · 15 Mar 2016

Replies to the reviewers' comments.

Reviewer 1

The paper focuses on whether the use of the fractional contribution of the m/z 44 signal to the total organic aerosol mass spectra (f44) can be used as an oxidation indicator. The authors have collected a certain amount of measurements of trace gases in order to draw their conclusions and make an educated guess for the missing information. The factor f44 was compared with the photochemical age of the pollutants (t[OH]) calculated from NOx/NOy, as the toluene/ethyne concentration ratio (NOx/NOy and hydrocarbon

clock, respectively) is not applicable in this situation and it was found to increase as t[OH] increased. This led to the conclusion that the factor f44 can be used as oxidation indicator and a discussion about the applicability of this method and possible causes of discrepancy with previous studies was included. The manuscript is in generally clear though I think some improvements in the quality of the figures are necessary. I would suggest considering moving most of the time series of trace gases to the SI or at least to create a more compact one page figure with all of them together. As they are now they do not really improve the quality of the paper or give such additional information that requires them to be plotted individually.

Authors' reply Thank you very much for your review. The similar comment on the time series plot (Figure 2 and Figure 3) has been raised by other reviewers. Due to the various magnitudes with many data points, combining all these time-series figures into one did not make the plots legible. We, therefore, decided to move the plots to the supplementary information (Figure S-4a to 4g).

One major point I have is the choice of the average OH concentration for the lifetime of CO, ethyne and NOy in section 3.3 and for the estimation of the competition of the O3 reaction in section 3.5. It is not clear how the authors arrive at the estimated concentration of average OH. The authors should clearly justify their choice, and as it is an estimate they should give a good idea of the effect of changing the estimation, i.e. the sensitivity of their estimate. Later in that section (lines 17, page16) it is reported that a value of 3 x 106 molecules cm-3 would be 6 times large than the concentration reported previously by Irei et al. 2014. As far as I can tell, there is no measured OH concentration reported in the cited paper.

Authors' reply The reason for the choice of average [OH] is given clearly in the revised manuscript (line 261, 388-389, 545). After the other reviewer pointed out, we realized that the discussion using 3x10ˆ6 molecules per cc of OH radical (the [OH] required to have the comparable reaction rate to that of R2 channel) was incorrect because of the slow reaction of N2O5 with water, the series of R2-R4 is more likely insignificant as the

NOx sink, unless otherwise R4 channel is significant, which is not supported by the acidity of aerosols observed.

In addition I think that figure 9 requires some additional explanation as it is hard to understand and as some suggestions for values of background mixing ratios of NMHC in the region of East China Sea are drawn from this figure. I would suggest including a legend to make it easier to immediately identify which color is which. Would it be possible to highlight the observed trend that on line 17 on page 18 is lying between the trends for the dilution with the background air and the reaction loss? Also, how can I see (line 10 on page 19) from figure 9 that the background NMHC ratios seem to lie between -3.5 and -4?

Authors' reply The legend was added to the updated figure (Figure 7). To impress the influence of vehicular emissions and natural gas, the sentences were re-organized (line 454-459). The suggested NMHC ratios of -3.5 and -4 at the background were meant to the lowest values of the natural logarithms of toluene/ethyne and i-pentane/ethyne ratios, respectively. The explanation was not sufficient, so the sentence was revised (line 474-476).

Specific comments: In general I would suggest to remove the "Note that" from the manuscript.

Authors' reply The majority of "note" was deleted from the text.

- Line 2, page 4: the wind is from west to east

Authors' reply Corrected (line 53).

- Lines 25 to 27 page 4: it is not totally clear what is given the evidence of SOA production

Authors' reply We meant that a systematic trend for the fractions of carboxylate in the organic aerosol (f44) with t[OH] is the evidence of SOA. The sentence was rephrased (line 35-36).

[Figure]

- Figure S2: I would suggest putting the cardinal points directly on the figure

Authors' reply The four directions were labelled with N, E, S, and W in the updated Figure S-2 and S-6.

- Table 1: what does n stand for?

Authors' reply It is the number of data points. The column label was revised in the updated Table 1.

- Lines 10 to 12 page 9: It is reported that the concentrations of organic aerosol reported by Irei at al. 2014 were relatively low. How does this fit with the current study?

Authors' reply The 10-day low concentration period Irei et al. (2014) studied is a part of the half-year study period reported in this paper, so the results here is more like the overall evaluation of the data from low concentrations to high concentrations. Your comment made us realize that in the text there should be a better place this information should be inserted in. The sentence was revised and inserted in line 189-191.

- Lines 10 to 13, page 11: It is not clear to me the meaning of this sentence. High mixing ratios occurred for a small portion of the observed data or high mixing influenced by local pollution occurred for a small portion of the observed data. Is it possible to distinguish between the two cases?

Authors' reply We have distinguished the pollution episodes of local and long-range transport origins by the durations of the episodes. That's what the sentence was supposed to mean. The sentence was rephrased to make this message more clearly (line 248-251).

- Line 1and 2 page 12: what is the meaning of recently improved emissions in NOy?

Authors' reply The other reviewer pointed out misinterpretation of discrepancy due to "improved emission of NOy". This discussion was thoroughly changed (line 273-282).

- Line 13, page 13: which data point?

Authors' reply "that data point" was changed to "the data point of f43 and f44" in the revised text (line 322).

- Line 28, page 13 to line 2, page 14: I suggest rephrasing the sentence.

Authors' reply The sentence was rephrased (line 336-339).

- Line 3 page 16: 1 x 106 molecules cm-3 of OH radicals do not correspond to 0.05 ppbv: : :rather to 5 x10-5 ppbv.

Authors' reply The unit was corrected to pptv for the concentration of OH radicals (line 386).

- I would suggest to separate in section 3.6 the two different t[OH] estimates with two subsections. It should also be stated in a clearer way that the estimate of t[OH] from the hydrocarbon clock is not possible for this conditions.

Authors' reply I understood that the section you referred is 3.5 (Chemical clocks). We divided the section into section 3.5.1 and 3.5.2. The discussion was revised to make the issue of mixing with background air clear (line 452-466).

Reviewer 2

General Comments: My understanding of the purpose of this paper was to determine the age of air masses based on gas and particle phase oxidation products transported downwind of a highly polluted region. This work is motivated by a desire to understand the strong and evolving influence that point sources of air pollution in China have on the surrounding regions. It is clear in this paper that a lot of experimental work was done. The analyses performed are explained well and are thorough. All of the results are presented well, but I feel conclusions drawn from the results could be improved. For instance, what does it mean to have a similar f44 increase rate at your site as was observed during the New England Air Quality Study? I'm having trouble bridging the conceptual gap with how determining the age of these transported air masses combined with the chemical oxidation product information gives us useful information

about SOA or transported air pollution generally. Having this explicitly and simply laid out would be very useful, but it is not necessary.

Authors' reply Thank you for evaluating our manuscript. The abstract was revised so that readers will have an clear idea how useful the information here is and what the agreement with the results from the other study implies for (line 39-44).

Specific and technical comments: section 3.3 – line 15 or so... I would change "highly correlated" to just "correlated"... that might be even stretching it with an R2 of 0.562 for OA but with ambient measurements "correlated" seems fair

Authors' reply "highly correlated" in the text was replaced with just "correlated" throughout the text.

section 3.3 – line 21 or so ... can you site a source for your average OH concentrations? generally to equate OH concentration into an equivalent "OH exposure day" in chamber studies we will use 1.5x10ËĘ6 [Mao, et al. 2008].

Authors' reply The $5 \times 10^5$ molecules per cc was from Takegawa et al. (2004), who estimated it from the hydrocarbon clock measured in the plume from the Asian continent and Japan. They also experimentally determined lifetime of NOy. We were supposed to refer the publication there, but did not. The reference was cited in the revised manuscript (line 261).

section 3.3 – "The high correlation (of particulate ammonium) with [delete organics] organics (m/z 44) suggests that (organics are primarily composed of carboxylic acids). [delete beyond here] in major the organics composed of carboxylic acids.

Authors' reply It's corrected (line 284-285).

section 3.3 – I didn't really notice before here, but ammonium (NH4+, which is measured by the AMS is referred to as "ammonia" here which is not correct)

Authors' reply Thanks for pointing this out. All "ammonia" was replaced with ammonium

(line 283-292).

section 3.4 – what are the correlation values between your extracted spectra from PMF and the spectral database? What are the correlation values between your December spectra and the spectra described in this study?

Authors' reply The coefficients of determination for the correlations between the LV-OOA and the reference LV-OOA from the database and between the LV-OOAs from this study and from Irei et al. (2014) were 0.94 and 0.98. Those for the HOAs (excluding the signals at m/z 27 and 29) were 0.53 and 0.98, respectively. The coefficients of determination were provided in line 313-317.

General comment – The term "signifigance" is used, generally, in scientific literature to describe a statistical significance. When "the significance of a reaction channel" is being evaluated you could alternitavely say "the relevance". You could also say something like "the reaction of x with y is dominant during the day" as opposed to significant.

Authors' reply According to your advice, "significant" and "insignificant" line 372 were replaced with important and negligible. "significant" in line 379 was replaced with dominant.

Page 21 - grammatical correction "This hypothesis consistently *explains our observations that the f44 oxidation indicator sometimes worked, and sometimes did not." Authors' reply It's corrected (line 527-528).

Reviewer 3

Overview: This manuscript presents measurements of mixing ratios of selected trace gases and composition of sub-micron secondary organic aerosol at a rural location in Southern Japan. The authors compare these measurements to previous measurements from this site and other sites in the region, discuss correlations between measured species, and estimate the photochemical age of air masses using concentration

ratios of selected trace gases. The stated purpose was to use these age estimates to explain variation in the concentration of secondary pollutants, specifically SOA and ozone. Understanding the evolution of SOA and factors influencing ozone formation are highly relevant areas of research especially in the southeast Asia region. The major strength of this manuscript is the high quality dataset generated at a site that receives air masses from industrialized areas of China after a short transit across the East China Sea. The experimental approach is straightforward, but sound, and the methods used are adequately described. The paper is, for the most part, well written with clear organization. Ultimately, the authors present an independent, quantitative estimate of the relationship between photochemical age and oxidized organic content of SOA; however, the manuscript does not clearly address the stated objective of determining the relationship between photochemical age and ozone mixing ratios. While this relationship is discussed in lines 362-368 and 385-392 and in Figure 10, the manuscript could benefit from a clearly identified and consolidated discussion of this relationship similar to that in section 3.6 for f44 vs. t[OH]. Alternatively, the mention of ozone in line 79 should be eliminated. In addition, several other issues should be addressed to improve the clarity of the manuscript as detailed below.

Authors' reply Thank you for your evaluation. Discussion on O3 mixing ratio and t[OH] is given in section 3.6 in the revised version. Therefore, the statement for the association of t[OH] with O3 was left as it is in the introduction (line 81).

Specific comments: Title: The term "oxidation products" could be more specific as many readers would consider this to imply that gas phase oxidation products (OVOCs) were measured. Perhaps mention SOA or use the term "secondary air pollutants".

Authors' reply If we used "secondary air pollutants", we would be using the term "pollutants" twice in the title. Instead of such repeat use, the title was revised to "Photochemical age of air pollutants, ozone, and SOA in transboundary air observed on Fukue Island, Nagasaki, Japan"

Abstract: It may be beneficial to mention the range of t[OH] calculated using NOx/NOy. Given the stated purpose of the study, some mention of the relationship between ozone and NOx/NOy should be included. Can the importance of the calculated f44 increase rate be put in better context in final sentence instead of a simple comparison to the NEAQS data?

Authors' reply The range of t[OH] was added in line 32-33. The relationship between O3 and tOH was added in line 40-43. The importance of information here is given in line 43-44.

Introduction: The sentence on lines 50-51 indicates that air masses move from east to west; this is opposite of the direction from China to Japan. Otherwise, this section is clear and concise.

Authors' reply The mistake was corrected (line 53).

Experimental: This section is also sufficiently thorough, but also concise. A more detailed description of potential local sources of trace gases in the study area would be beneficial for readers unfamiliar with Fukue Island. For example, are there agricultural operations that could contribute to the high particulate nitrate concentrations, or combustion sources other than automotive traffic that could contribute to ethyne and CO?

Authors' reply The site is located in-between the small pastures. There may be influence of vehicular emissions as local farmers mow the pastures by a tractor. This description was added in line 89-90. Other than this, home incinerator, agricultural waste burning, and traffic emission can be possible sources for CO, the NMHCs, and measured aerosol chemical species (line 90-92). We do not know other possibly important sources for these chemical species. We have not detected apparent influence of nitrate PM during the farmers' activity, such as a use of nitrate fertilizer.

Results and discussion: Section 3.1: Lines 139-140: How were precipitation events

determined to have a negligible effect on trace gas and aerosol data?

Authors' reply The line number you refer is probably 106-109 in the original manuscript submitted. We meant that the frequency of precipitation events was less enough that the overall results of evaluation including the data collected during the precipitation events do not change much. To avoid misunderstanding we rephrased to "Precipitation events were observed occasionally, but their frequency and strength did not seem to significantly affect our overall interpretation of the entire data set." (line 155-156).

Section 3.2: Lines 167-173: Are there agricultural operations in this region that couldãÅĂcontribute to the high particulate nitrate concentrations?

Authors' reply During the period of our study, we did not observe significant impact of these chemical species by agricultural operations.

Lines 224-228: The seven high-concentration episodes were derived from industrialized areas of the Asian continent. This text should be added because the clean air masses originating in Mongolia are transported from the Asian continent, too.

Authors' reply This part was criticized by the other reviewer. Although there were trajectories passing the industrialized area (i.e., south-west of Beijing), there were trajectories coming from other places. Back trajectories are a tool to provide a rough idea of air mass origins. This is stated in line 238-243 in the revised manuscript.

Section 3.3 Lines 242-243: Ethyne is a tracer for combustion sources in general, not just vehicular emissions.

Authors' reply The sentence was revised (line 257-258).

Line 250: Explain what variables were used in the regression. Was it CO and NOy?

Authors' reply Yes, the regression was meant for CO and NOy. It is clearly stated in the revised manuscript (line 265-268).

Lines 253-254: What do you mean by recently improved emission of NOy? Reduced

emissions?

Authors' reply This discussion was thoroughly revised (line 273-282). "improved" or reduced emission of NOx was incorrect.

Section 3.4 Lines 289-290: What was the match percentage for your HOA and LVOOA spectra compared to the Ulbrich database?

Authors' reply The rˆ2 for the correlation between the HOA and LV-OOA mass spectra in this study and the reference HOA and LV-OOA mass spectra from the database was 0.53 (excluding m/z 27 and 29 in this comparison due to influence of wide peak width of m/z 28) and 0.94, respectively. The information is given in line 314-316.

Lines 299-300: What does the high OM/OC ratios similar to humic-like substances say about the sources of your observed OA?

Authors' reply The high OM/OC ratios indicates that the organics was made of HULIS. However, this information cannot give any clue for its source information.

Section 3.5: Temperatures in this section are given in K and _C. Choose one unit and be consistent throughout the manuscript. Also, at some point in this section, it is important to explicitly state that no reliable t[OH] was calculated from the hydrocarbon clock.

Authors' reply Thanks for pointing out the mistake on temperature unit. It is corrected throughout the text. The limitation of t[OH] estimation by the hydrocarbon clock is stated in line 462-466 in the revised manuscript.

Lines 313-315: Consider rephrasing these sentences to indicate that because there are few potential sources of these gases between emission sources in Asia and the study site, this study offers an opportunity to use photochemical clock estimates under nearly ideal circumstances.

Authors' reply The sentences in line 341-343 meant so. Please let us know if we

misunderstood your suggestion.

Lines 374-376: The difference is consistent with what? Please clarify your meaning.

Authors' reply This part was removed from the text because the discussion using 3x10ˆ6 molecules per cc of OH radical (the [OH] required to have the comparable reaction rate to that of R2 channel) was incorrect due to the slow reaction of N2O5 with water.

Lines 446-447: These values refer to "natural log-transformed hydrocarbon ratios"

Authors' reply The sentence was revised (line 476-478).

Line 498: Significantly low what? It looks like something is missing in this sentence.

Authors' reply This part was revised to "a significantly low f44" (line 526). Actually it was already corrected in the manuscript published in ACPD.

Section 3.6: This discussion could benefit from a clear statement comparing the proportions of HOA and LV-OOA observed in the 2014 measurements (Irei et al. 2015) and in the current study and the correlation coefficient with t[OH] or NOx/NOy for each dataset. This may be helpful in determining a minimum proportion of LV-OOA necessary to use f44 as an indicator of oxidation.

Authors' reply You probably meant our publication in 2014 (Irei et al., 2014, EST). The comparison of PMF results between the previous and this study has been made in section 3.4 (line 310-328). Theoretically, there is no minimum proportion of LV-OOA to work as an oxidation indicator, but maximum proportion, at which f44 starts levelling off as t[OH] keeps increasing. This would depend on f44 values of two members. This explanation was added in line 522-528 in the revised manuscript.

Summary: Again, the use of the term "oxidation products" could be more specifically referred to as oxidized organic particulate matter, and some mention of the relationship between ozone and t[OH] should be made.

[Figure]

Authors' reply "the oxidation products" was rephrased to "the ozone and SOA formation from the oxidation of atmospheric pollutants". Some statements for the relationship between ozone and t[OH] were also added (line 572-575).

Figures: Figures 2 and 3: Can these be combined to include wind direction in Figure 1? If not, it would be helpful to have percentages on the wind rose in Figure 3 to indicate the distribution of wind direction observations.

Authors' reply Thanks for your suggestion, but it is not clear what percentages of. Do you mean percentages in frequency of occurrence above concentration thresholds arbitrary set? For your information, polar plot of chemical species concentrations are shown below. It can be seen that apparent sector-dependence was not observed in many chemical species, except NOx and m/z 57 fraction (f57): there is some sector-dependency in NOx and f57, but frequency was not many. For this reason, we did not go for further discussion on wind-sector dependence.

Figure. Wind-sector dependence of various chemical species concentrations.

Figure 5: This figure is quite large. Can panels with species with similar concentration ranges be combined? Also in panel (e), why does the baseline concentration of isopentane decrease after the break in the data? Was this a calibration issue?

Authors' reply I guess, you refer the time-series plot (Figure 3a-g). Based on your and the other reviewers' comments on this figure, we decided to move the figure to the supporting information (Figure S-4a-4g). Regarding the i-pentane issue, there was a chromatographic problem for the peaks of i-pentane around this period, and we, therefore, removed the suspicious data.

Figure 6: A boxplot of CO mixing ratios binned by wind direction may be more useful in demonstrating the lack of wind dependence. Or adding a mean or median line to the wind rose would help.

Authors' reply From the figure of polar plot of CO (Figure S4), it seems clear that the

statistics of wind sector dependency of CO mixing ratio would not provide a distinctive trend. To make it clear, box plots were created (see the figure below). In this plot, north (N), north-east (NE), east (E), south-east (SE), south (S), south-west (SW), west (W), and north-west (NW) were defined as the angle ranges of 337.6-22.5, 22.6-67.5, 67.6-112.5, 112.6-157.5, 157.6-202.5, 202.6-247.5, 247.6-292.5, 292.6-337.5 degrees, respectively, where 0 degree is defined as north. The standard deviations ranged from 80 to 120 pbbv, so we would say that the variation of the medians are insignificant.

Fig. Box plot of CO mixing ratio depending on wind direction. The horizontal bars inside the boxes, the upper and lower horizontal bars of the boxes, the lower and upper whiskers attached to the boxes stand for median, upper quartile, lower quartile, maximum, and minimum values, respectively. Standard deviations for the dataset of each direction ranged from 80 to 120 ppbv.

Figure 12: It would be helpful to show an overall trendline for the data to allow for a comparison with the modeled trends.

Authors' reply I guess, the reviewer is referring to the f44 plot against t[OH] (i.e., Figure 10 in ACPD version). The linear regression shown in the figure is the overall linear regression. Because the range of t[OH] was not so wide, we thought that the linear regression would be the best to evaluate the observed overall trend with minimum personal bias.

Tables: Double check that consistent significant figures are used in all tables. In Table 1, for example, the Max CO mixing ratio is given to 4 significant figures, the median to 3 sig figs and the minimum to 2 sig figs.

Authors' reply A significant figure depends on the digit where uncertainty exists. Based on the detection limit of CO, 10 ppbv, the uncertainty is in the order of a few ppbv. That is why the values of CO in Table 1 are given in different significant figures. However, the digits for $O_3$ and NMHCs seem to be inconsistent. The digits for the mixing ratios of these chemical species in Table 1 were corrected. In Table 3 the OM/OC ratios were

rounded to one decimal point. Thanks again for your review.

Reviewer 4

Review of "Photochemical age of pollutants and oxidation products in transboundary air observed on Fukue Island, Nagasaki, Japan" for Atmospheric Chemistry and Physics The authors have collected an interesting data set of trace gas and aerosol observations from a site in Japan which is exposed to continental outflow from the Chinese mainland. The title leads with "Photochemical age". Figure 10 based on NOx/NOy shows a reasonable trend in that there is more ozone in older air masses. There is a link between photochemical age and f44, though very noisy. An apparent conflict with the authors earlier work is examined with a model that gives f44 in terms of the properties of HOA and LVOOA, the amounts and properties of which differed between campaigns. A parameterization is arrived at with multiple constants for fitting, some of which may be derivable. That aspect deserves discussion. Unfortunately the differences between campaigns is not fully resolved. In regard to the trajectory analysis I recognize that the accuracy of individual trajectories is generally not high enough to make definitive statements. When considered in groups one can gain insights as to source information. I believe that the source identification would be more persuasive if the experimental period were divided into sets with 1) episode levels of CO and 2) mid or low levels of CO and the ensemble of trajectories for these conditions compared. In regard to photochemical age: There are many ways in which ratios can give biased age. In parts of this paper photochemical age is treated as having quantitative potential, as in the discussion of rate constant for OH+NO2. But in the end the authors seem to get it right, a valuable tools to give information on the relative effects of atmospheric processing Some of my comments are ways in which this manuscript could be improved. Some are critical. There are conclusions, which may or may not be correct, but are not substantiated. There are serious errors concerning nighttime chemistry of NOy and the lifetime of NOy which need to be corrected before this manuscript is considered for publication.

Authors' reply Thank you for spending your precious time to evaluate our manuscript.

Comments Abstract, line 9-11 correlation of NOy with CO implies negligible loss of NOy by reaction with OH radical. Loss from the NOy family of compounds due to reaction with OH is very slow. Loss of NOy is mainly from wet and dry deposition. Oxidation of NO2 to form HNO3 is rapid, but does not lead to loss of NOy. HNO3 so formed could partition to the aerosol phase but still would be detected as NOy by a NOx dectertor with a heated Mo catalyst.

Authors' reply Thank you for your critical, but constructive comment. We have neglected the wet and dry deposition of HNO3. The importance of wet/dry deposition caused major revision in the discussion, but the conclusion based on the observations remained the same. We considerably revised for this discussion. The details of revision can be found below.

Abstract, line 21. the f44 increase rate was 1.05(-9) hËĘ(-1) moleculeËĘ(-1) cmËĘ(-3). The terminology is confusing. f44 is dimensionless and its rate of change should have units of 1/time. The f44 increase rate is actually given by 1.05(-9) hËĘ(-1) moleculeËĘ(-1) cmËĘ(-3) [OH], where [OH] has units of molecules/cmËĘ3. The presentation in Eq. 3 is correct.

Authors' reply We agree with your point, the terminology issue. All "increase rate" are now given as the slope times [OH] (line 37, 43, 502, 543).

Page 5 Discussion of inlets and NOy measurement. Given the inlet dimensions and flow rate, there is approximately a 30 second transit time from the top of the inlet to the NOy instrument. Unlikely that HNO3 would make it through. A description of the measurement used in the present study should include the converter location

Authors' reply A statement for the location of molybdenum converter was added in line 97-98.

Page 11, line 15-17 CO was highly correlated with NOy (rËĘ2 = 0.674), ethyne (rËĘ2

= 0.724) and organic aerosols (rËĘ2 = 0.562) These correlations are not that high in comparison to observations in other studies. I do not know if the modifier "highly" is warranted. The present data set is accumulated over a few seasons and at a location that has long range transport from multiple directions. The diversity of emission sources and degrees of atmospheric processing will suppress the overall correlation.

Authors' reply "highly" was removed (line 255 in the revised manuscript).

Page 11, lines 20-21 at an average OH of 5e(5) the lifetime of NOy is 1.7 days. This is incorrect. 1.7 days is the right order of magnitude for oxidation of NO2 by OH under the assumed low OH concentration. Oxidation of NO2 transforms one member of the NOy family to another. It does not change the NOy concentration. The lifetime of NOy is set by wet and dry deposition and is affected by OH only in so far as reaction changes wet and dry deposition rates.

Authors' reply Thanks for pointing this out. The statements were incorrect, and we revised the statements to compare the lifetimes of CO, ethyne, and NOy (line 258-265). 1.7 day lifetime was adopted from Takegawa et al. (2014), who experimentally determined NOy sink during an aircraft campaign over Japan.

Page 11, line 23-25 NOy and CO were transported over a short distance This conclusion relies on a 1.7 day lifetime for NOy and is therefore suspect.

Authors' reply The lifetime seems to be able to explain the difference between the observed NOy/CO and the NOx/CO at emission. However, we admit that the expression of "short distance" was inappropriate, thus removed.

Page 11 – 12. Discussion of NOy to CO ratio. The observed ratio from a linear regression is 0.03. A study published in 2002 gave a ratio of 0.1 and a model calculation published in 2012, for air masses which had been transported long distances, gave a ratio of 0.03. The change in the ratio from the 2002 study to that from the present paper and the 2012 study is ascribed to recently improved emission of NOy. The text

implies that these few scattered numbers are due to emission changes. If that claim is made it should be documented. In the U.S., emission controls have caused urban ratios of NOx to CO to increase, at least up to around 2010 (see work by Parrish et al). Low NOy/CO ratios of order 0.03 are usually indicative of long range transport in which NOy has been partially removed by deposition. A minor point: One usually does not refer to NOy emission but rather NOx emissions.

Authors' reply Thank you for the critical and constructive comment. We found that the NOy/CO ratio of 0.1 Takegawa et al. reported in the text does not match with their actual observations shown in the figure. Their figure rather shows the ratio of 0.38 for the plume originated from Japan. So the NOy/CO ratios from the independent three studies are in the same order. The 1.7 day lifetime of NOy Takegawa et al. reported is based on their aircraft observations. It is not due to the reaction with OH, but the deposition. Given the deposition as the major sinking channel of NOy, the lifetime of NOy in our study is expected to be the similar order, unless otherwise the wet deposition, which we neglected, were significant in their study. Meanwhile, the back trajectories showed the transport time from the Chinese coast to our measurement site was roughly between one and two days. Considering the lifetime and the rough transport time, we admit your point that NOy likely sank partially. According to Kurokawa et al. (2013), the NOx/CO ratios at industrial emission in China are higher than 0.05. This supports your point as well. In addition, Kurokawa et al. report that the emission of NOx and CO in China also kept increasing from 2000 to 2010. This contradicts to our previous statement of "reduced NOy (corrected to NOx in the revised manuscript) emissions". Therefore, partial sink (possibly ∼50% or more) of NOy more likely explains our observations and other reports consistently. The sink is more likely the wet/dry deposition of HNO3. Nevertheless of the significant partial sink, the correlation of 0.67 between CO and NOy, the chemical species with the significantly different atmospheric lifetimes, and the better correlation between CO and ethyne (the chemical species with the longer lifetime than NOy) imply that the correlations are associated with the lifetimes in some extent. Consistent results of NOy/CO ratio by Takegawa et

al in their air craft measurements with our half-year ground-based observations under the low frequencies of precipitation events likely suggest that the sink is mainly due to the dry deposition. This discussion was revised (line 254-282). By the way, we could find a following publication by Parrish et al. (2010): Impact of transported background ozone inflow on summertime air quality in a California ozone exceedance area. This reference does not seem to be right one. We appreciate if the reviewer inform us the source of publication more specifically.

Page 11, line 4-5 and following. high correlation of particulate organics and ammonium suggest that in major the organics composed of carboxylic acids In my opinion this conclusion, true or not, is not demonstrated by the data provided. An air mass that comes from a polluted region is likely to have high concentrations of multiple pollutants. Thus a correlation between organics and NH4 could be due to 1) a correlation between sulfate and organics and 2) a correlation between sulfate and ammonium. It is difficult to disentangle multiple interactions. One could start with multi-variable regressions or PMF calculations that include inorganic ions.

Authors' reply The coefficients of determination given in Table 2 exhibit that the $r^2$ of 0.639 between $NH_4+$ and $SO_4^{2-}$, of 0.430 between $SO_4^{2-}$ and organics, and of 0.696 between $NH_4+$ and organics. If your thought had been the case, I expect that the $r^2$ between $SO_4^{2-}$ and organics would have been higher than 0.6. In addition, the PMF analysis resulted in that LV-OOA, organic acid, was the major component. Furthermore, m/z 44, a marker for LV-OOA, had the highest correlation with $NH_4+$ ($r^2=0.755$). For these reason, we think our conclusion is consistent. The lower $r^2$ between $NH_4+$ and organics is due to some contribution of primary organics (i.e., organics represented by HOA in the PMF analysis).

Page 15, line 7 to Page 16, line 6 Paragraph on NO3 chemistry. The gas phase reaction of N2O5 with H2O is very slow. Removal of N2O5 by reaction with water occurs in aerosol. It is rapid for acidic aerosol but slow for near-neutral aerosol (Brown et al, Science, 311, 67-70, 6 Jan 2006; Zaveri et al, JGR 115, D12304, 2010). If the N2O5

is not removed, NOx will be regenerated. NO3 can also react with VOCs, primarily olefins. Depending on conditions, nighttime chemistry can remove most or almost no NOx. The reaction rate cited for NO2 + O3 is just the one way flux through one of a series of reactions.

Authors' reply We agree with your opinion. The bottom line of this discussion is that (1) the high correlation between O3 and the extent of NOx conversion to NOy indicates the night time chemistry of O3 with NOx was not an important channel for the NOx conversion, and (2) the high correlation is reasonably explained by the daytime photochemistry of NOx. Speculation of the minimum [OH] ($3\times10^6$ molecules cm$^{-3}$) was made under the assumption that the night time and daytime chemistry compete. However, the night time chemistry may not compete unless otherwise N2O5 was removed to the aqueous phase. Indeed, our observation demonstrate that the molar ratio of NH4/SO4 is approximately 3, suggesting that there was enough NH4 to neutralize SO4. This in turn imply that the updake of N2O5 by aqueous phase is very small, as you say. Therefore, we removed the discussion for the speculation of the minimum [OH] from the text (from the section 3.3).

Page 16, line 7-20 Relative effects of nighttime and daytime NOx chemistry on O3. The effects of daytime chemistry have to be considered. Photochemical O3 production in the day is a chain reaction creating several O3 for each NOx removed. By taking differences between the 75th and 25th percentile data in Table 1 (best I could do with data on hand) one obtains a qualitative estimate of 7.5 molecules of Ox produced per molecule of NOx oxidized. The actual value is lower by an unknown amount because of NOy deposition. Nighttime chemistry is less efficient in using NOx to remove ozone. If the same amount of NOx is removed in the day and night, there will still be a positive correlation between O3 increase and NOx decrease.

Authors' reply We understood your point that N2O5 formed in nighttime still contributes to the daytime photochemistry, unless otherwise N2O5 was taken up by aqueous phase. The plot of ln[NOx]/[NOy] vs [O3] (Figure 5 in the revised manuscript), however,

is made with hourly data including daytime and nighttime. If the nighttime chemistry (R2 channel) took place, we expect that the conversion would reflect to the plot. We, therefore, interpreted the plot that the nighttime chemistry was negligible under the condition of our field measurements.

Page 19, Section 3.6 Dependence of f44 on t[OH] There is some increase in f44 with age. Perhaps this would show up better if the data was binned or lowess fitted. It is puzzling that different studies gave different results. Eq. 4 makes sense. I don't understand why a and b are arbitrary parameters. What must be measured to get their values?

Authors' reply The a and b values in eq (4) are parameters determining the relative magnitudes of HOA and LV-OOA in the binary mixture, respectively. If the HOA and the precursor of LV-OOA (or SOA) were from the same emission with the constant ratio, we expect that the same a and b values will be observed in different studies. If sources of HOA and the precursor are different, those values would vary, depending on emission strengths, dilution, and etc. By piling up more observations in different studies, we will be able evaluate whether or not those values vary largely or can be averaged out.

Minor points Page 5, Line 23-24. Could you please supply DL for NO and NO2. I am surprised to see a single figure for both as the measurement of NO2 is done by subtraction and involves the LED efficiency.

Authors' reply NO and NO2 were measured by the same chemiluminescent NOx analyzer, but the loading channels were different. So the DLs for NO and NO2 are the same (line 126-127).

Page 6, line 16-17 Are the AMS detection limits for a 10 minute period?

Authors' reply Yes, all AMS measurements here are 10 min averaged concentrations (line 105).

Page 6, line 23 and 25 What averaging times are used in specifying detection limits for NOx, NOy, O3, and CO?

Authors' reply The DL for the ozone analyzer was corrected to 3 ppbv. The averaging times used for determination of DLs for these chemical species were 1 min averaging time. The information of averaging time was added (line 127 and 129).

Page 14-15 Photochemical age, in particular choice of reaction rate constant for OH+NO2. The range of values due to temperature and pressure is small compared with other systematic errors such as the ratio of NO2 to NOx and the occurrence of reaction channels (e.g. PAN formation) that remove NO2.

Authors' reply In Table 1 the statistics for NO was added. The proportion of NO in NOx was very minor (the median of NO in NOx is smaller than 1%). This is stated in line 218-219 in the revised manuscript. Possible bias caused by formation of PAN is described in line 402-416.

Page 15, line 1-5 I am confused by the sensitivity calculation. As I understand, kNO2 is between 9.3e-12 and 1.1e-11. A nominal value of 1.0e-11 was used in the calculations. The product of kNO2 *t[OH] must remain constant as it is determined by a measured ratio of NOx to NOy. From Eq. (1) if 1.1e-11 is used in place of 1.0e-11, t[OH] decreases by 9%. Going the other way, t[OH] increases by 8%. The stated range in bias is different; -10% and 5%. A change in temperature of plus or minus 5K is brought up but evidently is not what is used in arriving at the kNO2 values in line 330. But more important why bother with the extended discussion of the temperature dependence of NO2+OH, when there are much more significant factors. Factors left out are the ratio of NO2 to NOx and the occurrence of other reaction such as PAN formation.

Authors' reply The discussion was requested by our colleagues. Apologies if the discussion on this issue irritated you. We meant that the comparison between the calculated t[OH] values using the temperature-dependent kNO2 and the fixed kNO2 (*kNO2) resulted in the difference ranging from -10% to +7% (correcting +5% to +7%), relative

to the t[OH] from the temperature-dependent kNO2. Referring the t[OH] with the use of *kNO2 as *t[OH], the relative bias is defined as

(t[OH]-*t[OH])/t[OH] = (*kNO2•kNO2- kNO2•kNO2) /*kNO2•kNO2 .

We hope that the revised sentences (line 362-364) makes this understand more easily.

Page 20, line 19 extent of reaction Needs a definition.

Authors' reply It was defined as "extent of reaction processing x for the LV-OOA precursor" in the revised manuscript (line 511-513).

Figure 9. Why is the f44 of HOA exactly zero? OA/OC for this PMF component has an O to C ratio approximately equal to one.

Authors' reply I'm not sure if we are understanding your point (O to C ratio) correctly, but the reason why f44 is zero is simply that the HOA component from the PMF analysis contained insignificant magnitude of signal at m/z 44. If you meant the OM/OC ratios, those ratios for HOA are zero (Table 1).

Supplement The text implies that Figures S1 to S-7 show trajectories for end of each episode terminating in Pacific Ocean or Mongolia. This is a hard feature to pick out. In some cases (S-7) the last trajectory passes over the same regions as trajectories that are part of the episode, but with a greater wind speed. In Fig S2 the last trajectory terminates over the East China Sea. However, this trajectory is shorter than the others and appears to point toward the mainland.

Authors' reply Let us correct that S-2 and S-7 you are referring to are the episodes in Feb 11 (episode 2) and May 19 (episode 7) in Table S-1, respectively, which are S-6 and S-10 in the ACPD version. We also realized that the length of back trajectories in previous Figure S-5 to Figure S-10 was inconsistent, so the figures were updated so that legible trajectories with consistent duration (48h back trajectories) were drawn. Overall, large changes in the trajectories were seen at the starts and/or the ends of each episode. It is thought that those are the transition of episodes. During the episode

2, the trajectories started from the north-eastern coastal region (the north-eastern industrial region) in China. Roughly speaking, the beginning and end of episode 2, the velocity of air mass changed, which are indications of transition of air mass origins. For the episode 3 and 7, such transition was observed in the middle of episode period. It is possible that different point sources influenced air quality. From the series of back trajectory plots, the information we gained was that the pollutants were likely derived from the Shanghai, north-eastern industrial, and Korean (near Seoul) industrial regions. Because the trajectories are not so accurate, it is hard to pin down where the sources are. The explanation was revised (line 239-244)

Table S1 The main text, line 223, promises qualitative information on the concentrations of other (non-CO) chemical species. I was expecting average or peak values, not check marks. The foot note to Table S1 specifies that the check marks are for observation of remarkably high concentrations without specifying what "remarkably high" means, either on an absolute basis or relative to the average or frequency distribution of the ensemble of measurements. Additional information needs to be added to Table S1.

Authors' reply We meant "remarkably high" as relative values (i.e., observations of concentration rise). So the concentrations are not necessarily high in absolute scale. The footnote of Table S-1 and the explanation in the text was revised (line 238-239).

TYPOs, wording Page 4, line 2 "east to west" Should be west to east

Authors' reply Corrected (line 53).

Page 9, line 1. emission sources of nitrate Should be emission sources of NOx.

Authors' reply We meant emission source of primary nitrate. The term "primary" was added to the sentence (line 186).

Page 11, line 5 in major the organics composed of carboxylic acids Suggest: organics are primarily composed of carboxylic acids

Authors' reply The phrase was corrected so (line 285-286).

Page 21, line 7 photoxidation toluene Suggest: toluene photoxidation

Authors' reply Corrected (line 522).

Page 21, line 8 ".. parameter, the 4 of which are determined by PMF analysis" Eliminate "the"

Authors' reply The article was removed (line 523).

Page 21, line 8 starting with "More progress.." Not a sentence.

Authors' reply It's corrected to "The greater extent of reaction processing proceeds, the greater LV-OOA contributes to the binary mixture of HOA and LV-OOA, each of which has significantly different f44 value." (line 524-526)

Page 21, line 10 "containing a significantly low continues to increase. Words are missing

Authors' reply A word was missing in the original file, but it was already corrected in the paper published in ACPD (P21, L11). Thanks for pointing this out anyway.

Page 39, Symbols on Figure 6. Colors for top two categories difficult to distinguish. I have normal color vision.

Authors' reply The colors of the plots and their sizes were changed so that readers can easily distinguish the plots (revised Figure 4).

Page 41, x-axis of Figure 8. Dates should be the same as used in other figures, i.e. Dec 1, Jan 1, Feb 1, etc.

Authors' reply The x-axis was corrected (revised Figure 6).

Figure 11 appears to be missing from last version that I downloaded. In a previous version it had a time axis that did not match others figures.

Authors' reply The reviewer may have referred to the time-series plot of calculated t[OH] by NOx/NOy clock. Your point sounds the same as the previous comment (inconsistent different time format). The inconsistency was corrected in the revised manuscript (revised Figure 6). Please let us know if we misunderstood your comment. Thank you very much for your help to improve the manuscript.

---

## Editor Decision (ED1)

**Comments by Reviewer #4:**

(Page/Line numbers refer to the previous version of the paper; they are simply copied from the reviewer report)

Page 11-12:

I think the reviewer may have referred to studies by Parrish et al., such as Parrish, J. Geophys. Res. 2002 (10.1029/2001JD000720), or Atmos. Environ. 40 (2006) 2288–2300

Page 11, line 4-5 and following:

I think your response was great and includes important information; however, I did not find if/where you added it to the revised manuscript.

Figure 9:

"…If you meant the OM/OC ratios, those ratios for HOA are zero (Table 1)"

I got confused about your response here. 1) I don't see any OM/OC ratios in Table 1,

2) How can an OM/OC ratio be 0 (i.e. OM = 0?)? Do you mean for very low oxidized material OM/OC ~ 1?

**Editor comments**

Here the line numbers refer to your revised manuscript.

**Comments:**

l. 220: Can you give numbers for the LDL values?  Has LDL been defined before?

l. 265/6: I do not understand the argument 'Because the processes are physical we expect that the order of NOy lifetime in our study is similar."

l. 330: Can you add a reference to the statement that humic-like substances have high OM/OC ratios?

l. 383: Not clear. R4 as it is written does not include any pH dependence. Why does the fact that sulfate was neutralized by ammonia makes R4 negligible?

l. 572: Add the value and reference for the rate from the New England Air Quality Study here

l. 530: Not clear what you mean

"**As** two members, HOA and LV-OOA, had similar f44 values, the indicator **did** not work" or

"**If** two members, HOA and LV-OOA, had similar f44 values, the indicator **would** not work"

**Technical comments**

l. 93: sometimes

l. 128; 130 min: under 1 min averaging time → with 1 min averaging time

l. 144: 2,5 → 2.5

l. 152: 'standard deviation' can be omitted as SD has been defined before

l. 258: originate → originates

l. 268: in the similar order to → on the similar order of

l. 279: "Nevertheless of such… " – do you mean "Despite of…"?

l. 282: sinking → sink

l. 327: at some extent → to some extent

l. 382: remind → remember

l. 393: 4 times as fast as → four times faster than

l. 424: Robert → Roberts

l. 437: under → at

l. 507: another → other

l. 524: 'The greater extent of reaction processing proceeds..' – please reword

l. 573: implying → imply

l. 575: increasing → increase

Figure 7, l. 860: as function → as a function

---

## Author Response (AR2)

**Comments by Reviewer #4:**

(Page/Line numbers refer to the previous version of the paper; they are simply copied from the reviewer report)

Page 11-12:

I think the reviewer may have referred to studies by Parrish et al., such as Parrish, J. Geophys. Res. 2002 (10.1029/2001JD000720), or Atmos. Environ. 40 (2006) 2288–2300

**Author's reply:**

**The reference of Parrish et al. (2002) was added in line 276. Thanks for the information.**

Page 11, line 4-5 and following:

I think your response was great and includes important information; however, I did not find if/where you added it to the revised manuscript.

**Author's reply:**

**Apologies for the insufficient explanation. It is feasible to explain that the majority of organics was made of carboxylic acid that was fully or partially neutralized by ammonium because 1) there was more than enough $NH_4^+$ to neutralize sulfate and nitrate (stated in line 295-299), 2) the $r^2$ between m/z 44 and $NH_4^+$ was higher than the $r^2$ between m/z 44 (or organics) and sulfate (Table 2), and 3) the PMF analysis showed that LV-OOA, which can be estimated according to the marker of m/z 44, is the major component of organics (line 341-342). We revised the discussion (line 289-305) because your 2$^{nd}$ comment made us realize that our explanation for the neutralization by ammonium was confusing. Thanks for your comment again.**

Figure 9:

"…If you meant the OM/OC ratios, those ratios for HOA are zero (Table 1)"

I got confused about your response here. 1) I don't see any OM/OC ratios in Table 1, 2) How can an OM/OC ratio be 0 (i.e. OM = 0?)? Do you mean for very low oxidized material OM/OC ~ 1?

**Author's reply:**

**Figure 9 you refer would be Figure 8 in the revised manuscript. Apologies for**

referring the incorrect table and explaining the OM/OC ratios incorrectly. The table was supposed to mean Table 3. The OM/OC should not be zero. Zero we meant was f44 for HOA component.

Back to your original critics, we double checked the OM/OC ratios for HOA and LV-OOA components. We then found that the calculation for OM/OC was incorrect. Correct OM/OC ratios for HOA and LV-OOA are 1.7 and 4.2, respectively. The ratios in Table 3 and the text (line 353) were revised accordingly. This change also required a change in the parameter "a" in the model calculation. To have the same fit as the previous plot, the "a" value needed to change to 0.025. This change was made in Table 3 and line 559. Figure 8 was also updated accordingly.

**Editor comments**

Here the line numbers refer to your revised manuscript.

**Comments:**

l. 220: Can you give numbers for the LDL values? Has LDL been defined before?

**Author's reply: LDL for NOx and NOy means < 0.006 ppbv. "LDL, <0.006" was added in line 215.**

l. 265/6: I do not understand the argument 'Because the processes are physical we expect that the order of NOy lifetime in our study is similar."

**Author's reply:**

**The explanation was revised to give our point of view more specifically (line 266-271).**

l. 330: Can you add a reference to the statement that humic-like substances have high OM/OC ratios?

**Author's reply: A statement "Based on the AMS reference mass spectra available from the web site previously referred, …" was added in line 357-358.**

l. 383: Not clear. R4 as it is written does not include any pH dependence. Why does the fact that sulfate was neutralized by ammonia makes R4 negligible?

**Author's reply: Apologies for the poor explanation. The information is now given more specifically (line 406-410).**

l. 572: Add the value and reference for the rate from the New England Air Quality Study here

**Author's reply:**

**The rate, $1.3 \times 10^{-9} \times [OH]$ h$^{-1}$, was added (line 591).**

l. 530: Not clear what you mean

"**As** two members, HOA and LV-OOA, had similar f44 values, the indicator **did** not work" or

"**If** two members, HOA and LV-OOA, had similar f44 values, the indicator **would** not work"

**Author's reply:**

**Sorry for confusing. "HOA and LV-OOA" was removed because the "two members" are not necessarily HOA and LV-OOA. As you suggest, we revised the**

**sentence to as follows: "If two members had similar f44 values, the indicator would not work." (line 548-549).**

**Technical comments**

l. 93: sometimes

**Author's reply:**

**Corrected (line 93).**

l. 128; 130 min: under 1 min averaging time → with 1 min averaging time

**Author's reply:**

**Corrected (line 128 and 130).**

l. 144: 2,5 → 2.5

**Author's reply:**

**Corrected (line 144).**

l. 152: 'standard deviation' can be omitted as SD has been defined before

**Author's reply:**

**The term was removed (line 152).**

l. 258: originate → originates

**Author's reply:**

**Corrected (line 264).**

l. 268: in the similar order to → on the similar order of

**Author's reply:**

**Corrected (line 272).**

l. 279: "Nevertheless of such… " – do you mean "Despite of…"?

**Author's reply:**

**It was changed to "Despite" (line 284), to be consistent with usage of "Despite" in other places in the text.**

l. 282: sinking → sink

**Author's reply:**

**"sinking process" was changed to sink (line 287).**

l. 327: at some extent → to some extent

**Author's reply:**

**The sentence including "at some extent" was removed from line 355 because incorrect information, which was pointed out by the Reviewer #4, was presented.**

l. 382: remind → remember

**Author's reply:**

**"one should remind" was removed from the sentence (line 408).**

l. 393: 4 times as fast as → four times faster than

**Author's reply:**

**The sentence was removed due to the removal of the lifetime comparison (see the 5th in "Additional revision" below).**

l. 424: Robert → Roberts

**Author's reply:**

**Corrected (line 443).**

l. 437: under → at

**Author's reply:**

**Corrected (line 456).**

l. 507: another → other

**Author's reply:**

**Corrected (line 526).**

l. 524: 'The greater extent of reaction processing proceeds..' – please reword

**Author's reply:**

**"processing" was removed (line 543).**

l. 573: implying → imply

**Author's reply:**

**Corrected (line 592).**

l. 575: increasing → increase

**Author's reply:**
**Corrected (line 594).**

Figure 7, l. 860: as function ➔   as a function
**Author's reply:**
**The figure captions for Figure 7 and 8 were corrected accordingly.**

**Additional revision**
1. **The tile was not revised in the last revised manuscript. The title was updated.**

2. **The averaging time and DL value provided for CO was found to be incorrect. These values are corrected in line 130-131 in the most updated manuscript.**

3. **We may have misunderstood the original comment from the reviewer #4, "**Figure 9. Why is the f44 of HOA exactly zero? OA/OC for this PMF component has an O to C ratio approximately equal to one.**" The O to C ratio of one seems to be the 43/44 ratio shown in Figure S-14. We looked into the results of PMF analysis and found a contribution of semi-volatile oxygenated organic aerosol (SV-OOA) sometimes. The 43/44 ratio of one can be explained by a combination of HOA (or SV-OOA) and LV-OOA. This discussion was revised in line 331-338.**

4. **An additional figure supporting the additional statement for PMF analysis was added to the supporting information (Figure S-15).**

5. **Comparison of lifetimes calculated for the R1 and R2 channels were removed (line 384-394 in the previous revision) because the discussion was found to be redundant and meaningless after stating that the series of R2-R4 channels was negligible based on the clue by neutralization of sulfate and nitrate by ammonium. The discussion for the negligible nocturnal sink of NOx was revised (line 406-422).**

6. **Figure 2 and 3 were replaced with more legible ones.**